

# Technical note: Emission mapping of key sectors in Ho Chi Minh city, Vietnam using satellite derived urban land-use data

Trang Thi Quynh Nguyen [1,2], Wataru Takeuchi [2] and Prakhar Misra [3]

[1]   Space Technology Institute, Vietnam; ntqtrang@sti.vast.vn
[2]   Institute of Industrial Science, The University of Tokyo, Japan; wataru@iis.u-tokyo.ac.jp
[3]   Research Institute of Humanity and Nature, Kyoto, Japan; mprakhar@chikyu.ac.jp

*Correspondence to*: Nguyen Thi Quynh Trang (ntqtrang@sti.vast.vn)

**Abstract.** Emission inventories are important for both simulating pollutant concentrations and designing emission mitigation policies. Ho Chi Minh city (HCMC) is the biggest city in Vietnam but lacks of an updated spatial emission inventory. In this study, we propose a new approach to update and improve a comprehensive spatial emission inventory for major Short lived climate pollutants (SLCP) and Green house gases (GHG) ($SO_2$, $NO_x$, CO, NMVOC, $PM_{10}$, $PM_{2.5}$, BC, OC, $NH_3$, $CH_4$, $N_2O$, and $CO_2$) Our originality is the use of satellite derived urban land-use morphological maps which allow spatial disaggregation of emissions. Based on this approach, a comparable and consistent local emission inventory (EI) for HCMC has been prepared, including three key sectors as a successor of previous EIs. It provides annual emissions of transportation, manufacturing industries and construction and residential sectors at 1km resolution. The target years are from 2009 to 2016. We consider both Scope 1 - all direct emissions from the activities occurring within the city and Scope 2 that is indirect emissions from electricity purchased. Transportation sector was found to be the most dominant emission sector in HCMC followed by Manufacturing industries, and Residential area, responsible for over 682 Gg CO, 84.8 Gg $NO_x$, 20.4 Gg $PM_{10}$ and 22000Gg $CO_2$ emitted in 2016. Due to sharp rise in vehicle population, CO, $NO_x$, $SO_2$ and $CO_2$ traffic emissions show increases of 80%, 160%, 150% and 103% respectively between 2009 and 2016. Among five vehicle types, motorcycle contributed around 95% to total CO emission, 14% to total $NO_x$ emission and 50-60% to $CO_2$ emission. Heavy vehicles are the biggest emission source of $NO_x$, $SO_2$ and PM while personal cars are the largest contributors to NMVOC and $CO_2$. Electricity consumption accounts for the majority of emissions from Manufacturing industry and Residential sectors. We also found that Scope 2 emissions from manufacturing industry and residential areas in 2016 increased by 87% and 45% respectively in comparison with 2009. Spatial emission disaggregation reveals that emission hotspots are found in central business districts like Quan 1, Quan 4 and Quan 7, where emissions can be over 1900 times of ones estimated for sub urban HCMC. Our estimates show relative agreement with several local inherent EIs, in terms of total amount of emission and sharing ratio among elements of EI. However, the big gap was observed when comparing with REASv2.1, a regional EI, which mainly applied national statistical data. This publication provides not only the approach for updating and improving local EI but also the novel method of spatial allocation of emissions in city scale using available data sources.

## 1. Introduction



Emission inventories (EI) are key for identifying the source of pollutants. This is particularly true in South East Asia, where
the rise of energy demands results in significant air quality and human health issues. A number of regional anthropogenic EIs
exist to be used as input for atmospheric chemistry models and also to understand the long-term trends of emission level in
this area (Tab. 1). Many atmospheric chemistry modelling researches in Asia have applied these EIs as input data but they
are incoherent and not longer updated. Also, only a few attempts have been made to understand the annual evolution of
Asian emissions. REAS (Regional Emission inventory in ASia) is the first inventory to integrate time series of emission data
for Asia on the basis of a consistent methodology. REASv2.1 was developed from REASv1.1 with the spatial resolution of
gridded data improved to 0.25◦ × 0.25◦, and temporal resolution increased to monthly (J. Kurokawa et al., 2013) REASv3.1
was updated till 2015 and covers the longer historical time span from 1950-2015 (J. Kurokawa et al., 2019) The inventories
mentioned above were compiled in regional scale with coarse resolutions. They mainly applied national energy consumption
data as activity data. Apart from countries having their own database of emission factors (EF) like China, Japan and Taiwan,
EFs of the rest of Asian countries were extracted from many sources, including previous Asian EIs and recent studies. With
respect to grid allocation, the spatial distributions of rural, urban and national populations and road networks were used to
allocation emissions from area sources to grid cells. Obviously, these approaches are not suitable for community scale EIs
what demand higher detail levels of both activity data and spatial disaggregation.

According to Global protocol for Community-Scale Greenhouse Gas emission inventories (GPC), urban areas are
responsible for more than 70 percent of global energy-related carbon dioxide emissions and the achievement of emission
reduction of the economy in the upcoming decades will depend mainly on cities. Thus, it is very important to develop an EI
in city scale. At the same time, a continuous historical EI could show the long-term evolution of emissions as a consequence
of socio-economic development in cities. In response to these needs, the GPC establishes credible emissions accounting and
reporting practices that help cities to calculate and report on community scale greenhouse gases and develop their own
historical EI.

A research study by Yale University found that Vietnam's PM2.5 index ranked 170th out of 180 surveyed countries and this
is considered as one of the ten most polluted countries in the world in terms of air quality (Yale Univ.,2012) In urban area
like Hanoi and Ho Chi Minh city (HCMC), the situation has become worse because of high intensity of anthropogenic
activities. In 1st quarter 2018, the average PM 2.5 concentrations measured in Hanoi and HCMC reached 63.2 and 37.2
ug/m3, respectively (GreenID, 2018) However, while air pollution level in Hanoi exhibits strong seasonality and the
dependence of meteorological factors, air quality in HCMC in mainly influenced by anthropogenic emission occurring inside
the city (GreenID, 2018) For these reasons, in this study, we focus on the annual emissions of HCMC, Vietnam.

In 2017, the first comprehensive Green House Gases (GHG) Inventory of HCMC was prepared for 2013, 2014 and 2015
with the assistance of the Japan International Cooperation Agency (JICA) under the Project to Support the Planning and
Implementation of Nationally Appropriate Mitigation Actions in a Measurement, Reporting and Verification Manner (SPI-
NAMA). According to their calculation, among five main anthropogenic sectors, Transportation and Stationary energy are
two most prominent emission sectors in HCMC, comprising 45% and 46% of the total respectively. Within Stationary





energy sector, manufacturing industries accounts for the highest portion (46%), followed by Residential buildings (33%) (JICA, 2017) Another EI was compiled to calculate emissions in HCMC in 2017 and forecast for 2025 and 2030, including

on-road emission sources, non-road mobile sources, area source and biogenic sources by B.Q.Ho et al.., 2019. They collected coordinates and distribution of each source to prepare emission maps with 0.5 km resolution. For each pollutant (CO, NOx, SO2, TSP, NMVOC, and CH4), the emissions were calculated for each cell. In addition to these comprehensive studies, a several of EIs were developed for HCMC but mainly focused on road traffic emission (L.C. Belalcazar, 2009, B.Q.Ho, 2010; N.T.K.Oanh et al., 2015; L.T.P.Le et al., 2018) Only a few attempts have been made to spatial disaggregate

the emissions (B.Q.Ho, 2010) Besides, these studies have low level of consistency and inheritance from previous EIs. Also, with the rapid economic development in HCMC, the significant evolution of various emission sources is expected. Thus, it is important to compile a detail and continuous local EI for this city.

Besides, spatial distribution of emissions is a crucial step to fulfil the requirements of gridded EI as input data for air quality modelling. In conventional way, the spatial allocation of area source emissions is based on rural, urban and total population

data (J. Kurokawa et al, 2013; J. Kurokawa et al, 2019) However this method is not suitable to be applied in community scale EI, which requires high grid resolution. Especially, it is not rational to use population data for spatial disaggregation of Industrial sector.

In response to these needs, we developed the annual inventory, as successor of REASv2.1 and GHG emission inventory provided by JICA, focusing on three key sectors: (1) Transportation, (2) Manufacturing industries and (3) Residential

building, using higher detail level of activity data, local emission factors for HCMC and a novel approach for grid allocation. This local EI covers from 2009 to 2016 and includes emissions of following species: SO2, NOx, CO, non-methane volatile organic compounds (NMVOC), black carbon (BC), organic carbon (OC), CO2, NH3, CH4, and N2O, PM10 and PM2.5. In addition, both Scope 1 - direct emissions from the activities occurring within the city and Scope 2 that is indirect emissions from electricity purchased are considered.

Section 2 describes the methodology used in our EI to estimate emissions, including activity data, emission factors and spatial distribution of EI. Section 3, the results and discussion, covers four topics: (1) Emissions from each sectors; (2) Scope 1 and Scope 2 emissions; (3) Spatial distribution; and (4) Comparison with other inventories. A summary and conclusion are given in Sect. 4 and 5.

## 95  2. Methodology

### 2.1.    Study Location

HCMC is the most populous city in Vietnam with a population of 9 million as of 2019. We chose it over Hanoi capital because the updated EI with detail information about urban emission sources will have remarked significance for HCMC. Air quality in this city is mainly influenced by anthropogenic emission occurring inside the city. In other words, the relative

independence of situation in this city on other adjacent sources facilitates the compiling local EI. Meanwhile, air pollution



level in Hanoi exhibits strong seasonality and the dependence on meteorological factors. Also, the local emissions were not a dominant source of pollution (GreenID, 2018) Figure 1 shows the inventory domain of our EI.

## 2.2.     General description

Table 2 summarizes the general information of our EI that includes nine major air pollutants and three greenhouse gases, as a successor of REASv2.1: SO2, NOx, CO, non-methane volatile organic compounds (NMVOC), black carbon (BC), organic carbon (OC), CO2, NH3, CH4, N2O, PM10 and PM2.5. The target years are from 2009 to 2016, to continue the period covered by REASv2.1. Source categories considered in this inventory are basically the same with GHG emission inventory that compiled basing on the guideline of GPC. But here we focus on only three dominated sectors defined by GHG emission

inventory developed by JICA: (1) Transportation, (2) Manufacturing industries and (3) Residential building. The spatial resolution is improved to 1km to provide detailed grid nets for atmospheric chemistry models and emission maps for local government and decision makers. Besides, we collected more city-specific activity data and local emission factors (EF) from recent studies of emission inventories for Asian countries (see Sect. 2.3)

**2.3. Basic methodology**

**2.3.1. Transportation emission**

Figure 2a shows the flow of diagram for estimating emissions from Road transport. On road vehicles were classified as motorcycle (MC), taxi, car, bus and heavy duty vehicle – truck, each of which includes gasoline and diesel vehicles. We calculated annual hot emissions based on annual number of registered vehicles, average daily vehicle mileage traveled, and

emission factors for each vehicle type with the following equation:

$E_{hot} = \sum_i VP_i * DailyVKT_i * 365 * EF_i$  (1)

Where i represents vehicles types; daily VKT is the average daily vehicle kilometre traveled of vehicle type i (km/ day); $VP_i$ is the population of vehicle type i; $EF_i$ is hot emission factor of vehicle type i. The daily VKT of each vehicle type in HCMC, 2013 were extracted from study of N.T.K. Oanh, 2015 and was assumed to be constant over years (Tab. 3):

The vehicle population data were synthesized by different data sources, such as the statistic of The Transport Department of HCMC and previous studies about vehicle emission in HCMC. However, the annual number of registered vehicles in some types were missing, such as population of truck and bus over years. The number of trucks was calculated basing on the data in 2013 (N.T.K.Oanh, 2015), and proportionally estimated for other years basing on annual volume of freight carried that were provided by HCMC Statistical Yearbook. The bus population and the taxi population in 2015, 2016 were proportional

with number of cars (Tab. 4)

Cold emissions from road transport were included for NOx, CO, PM10, PM2.5, BC, OC, and NMVOC by the following equation:

$E_{cold} = E_{hot} * \beta_i * F_i$ (2)



Cold emissions were adjusted according to hot emissions using the fraction of distance traveled driven with a cold engine or
with the catalyst operating below the light-off temperature ($\beta_i$) and the correction factor of $EF_{hot}$ for cold start emissions ($F_i$)
The parameter β and F are functions of average monthly temperature. Equations for β and F and related parameters were
taken from the EMEP/EEA emission inventory guidebook 2009 (EEA, 2009). Monthly average surface temperatures in
HCMC were adopted from https://www.weather-atlas.com/

The emission factors were extracted from seven different studies conducted in Hanoi, China (Tab. 5), covering 12 pollutant
species. In Eq. (1), EFs and daily VKT of each vehicle type were assumed to be constant over years. Therefore, the annual
emission was mainly driven by vehicle populations.

### 2.3.2 Manufacturing industries emission

Figure 2b shows the basic procedure to estimate emissions from Manufacturing industry sector. It focuses on fuel-
consumption based emission which is considered as Scope 1 emission. Scope1 refers to all direct emissions from sources
located within the city boundary. Besides, this study will calculate Scope 2 - consumption-based emission separately, which
originates from electricity consumption (Sect. 2.3.4) Emissions from fuel combustion were calculated from the following
equation:

$$E_{Fuel} = \sum_i A_i * EF_i * (1 - R_i) \ (2)$$

Where E is emission from fuel consumption of Manufacturing industrial activities, i is fuel type, A is fuel consumption, $EF_i$
is unabated emission factor of each combustion species; $R_i$ is reduction efficiency of control device. In the case of SO2,
emission factor was estimated from the following equation:

$$EF_{SO2} = S_i * (1 - SR_i) \ (3)$$

Where $EF_{SO2}$ is emission of SO2 for each fuel type, $S_i$ is sulfur content of fuel, SR is sulfur retention in ash. The total fuel
consumption in HCMC in 2013, 2014 and 2015 with Ratio of Final Fuel Consumption by Sub-Sector and Fuel Type (Tab. 6)
were provided in the GHG emission inventory compiled by JICA, 2017. Basing on this GHG emission inventory, the annual
fuel consumption of Manufacturing industrial and Residential sectors, including gasoline, diesel, heavy oil, kerosene,
liquefied petroleum gas (LPG) and natural gas can be estimated for three years: 2013, 2014 and 2015. The fuel consumptions
of Manufacturing industrial sector in five other years (2009 to 2012 and 2016) were inferred using annual Gross output of
industry at current prices by industry activity in HCMC, provided by HCMC Statistical Yearbook (Tab.6) Unabated
emission factors, reduction efficiencies of each pollutant species, sulfur content of fuel, sulfur retention in ash were adopted
from the compiled database presented in the Atmospheric Brown Cloud - Emission Inventory Manual (ABCEIM) by
Shrestha et al.., 2013) ABCEIM has included EFs from several databases including the AP-42 (USEPA, 1995), EMEP/
CORINAIR (2006) and IPCC (1997), as well as available measurement data reported for various sources in Asia (Tab. 7)

### 2.3.3 Residential emission

Figure 2b shows the flow of diagram for estimating emissions from Residential sector. This sector covers all fuel combustion
activities in households, including domestic cooking and use of fireplaces. Kerosene, liquefied petroleum gas (LPG) and





natural gas are used for cooking, while kerosene is used for lighting in the residential sector in many regions. Coal, biomass fuels, such as wood are used mostly for domestic cooking and heating stoves in rural. Similar to Manufacturing industry
sector, the annual emission from Residential sector was calculated using Eq. 2 and 3. Fuel consumption in 2013, 2014, 2015 were provided by GHG emission inventory compiled by JICA, 2017 (Tab. 4) The fuel consumptions in other years were inferred using population provided by HCMC Statistical Yearbook (Tab.8)

The uncontrolled emission factors and reduction efficiencies for Residential sector, sulfur content of fuel and sulfur retention in ash were adopted from ABCEIM by Shrestha et al.. 2013 (Tab.5)


### 2.3.4    Emission from electricity consumption

Apart from Scope 1 emission, this study also considered CO2 Scope 2 emission that is from purchased energy generated upstream from the city, mainly electricity. Consumption-based emissions encompass those emissions produced by consumption within those same boundaries, regardless of the origin of those emissions. Local governments often include
Scope 2 emissions when they do not have electric generating plants within their boundaries but still wish to evaluate the impacts of electricity use in the community. CO2 emission from electricity consumptions of Manufacturing industries and Residential sectors were calculated from the simple equation:

$$E_{Electricity} = \sum \ A \times EF_{electricity} (4)$$

Where E is emissions from electricity consumption, A is activity data, here is amount of electricity consumption from each
sector, $EF_{electricity}$ is Grid Emission Factor, specific for each region. In GHG emission inventory compiled by JICA, the electricity consumption in 2013, 2014, 2015 by sub-sectors was collected from Electricity of Vietnam (EVN) using the data collection forms. The electricity consumption consists of five sub-sectors (Residential Buildings; Commercial and Institutional Buildings and Facilities; Manufacturing Industries and Construction; Energy Industries and Agriculture, Forestry and Fishing Activities) (Tab. 8) This value includes emissions from both Consumption of grid Supplied Energy
Consumed within the city boundary and Transmission and distribution Loss from grid Supplied Energy.

The significant linear relationships during 2013-2015 between electricity consumption of Industry sector and Residential sector with Annual Gross output of industry and Annual population, respectively were found (Fig. S1). So, electricity consumptions in other years were inferred using the same parameters used in Fuel consumption part:

-        Manufacturing Industries and Construction sector:  used Annual Gross output of industry at current prices by
industry activity.

-        Residential: used Annual population of HCMC.

The EF on electricity consumption varies every year. EFs depend on: combustion technology; emission source category; fuel type; combustion technology type and emission control technology. In GHG emission inventory of JICA, Grid Emission Factor on Electricity Consumption in Vietnam were provided for 2013, 2014 and 2015 (Tab 9) As a result, in this study, EF
on Electricity Consumption in 2009 to 2012 was assumed to be the same with 2013 and EF on Electricity Consumption in 2016 was assumed to be the same with 2015.



### 2.3.5.    Spatial allocation

Regarding on road transportation sector, in order to make grid emissions, the road density from road network downloaded
from Open Street Map (OSM [insert cite]) was applied for spatial disaggregation. In which, the grid net was created and road
density was estimated for each "square", with different weights for three types of roads: 2 for Primary roads, 1 for Secondary
roads and 0.5 for Tertiary roads. These weights were derived from Modelled road capacity in HCMC in 2016 by
HOUTRANS project, JICA, 2004. In which, the assigned traffic volume in Primary road is over 85,000 Passenger Car Unit
(PCU) per day, Secondary one is 44,000 to 85,000 and the smallest road have under 44,000 PCU per day (JICA, 2004)

To spatially allocate Scope 1 urban emissions from manufacturing industries, commercial places and residences, annual
urban land-use morphology map for period 2009 - 2016 in HCMC was created following Misra P. et al. (2019). Urban
morphology maps include three land use types: residential, commercial and industrial land. Urban morphological maps were
prepared at 30 meter spatial resolution by classifying digital building heights and nighttime light over each pixel into the
three land-use types.

Digital building heights were extracted from publicly available AW3D digital surface model (DSM) data at 30 meter
resolution. A DSM is a representation of visible geological earth terrain and any other features (tree and crop vegetation,
built structures, etc.) occurring over the ground terrain.

AW3D DSM data can be publicly obtained (since March 2016) at 1 second (30m) horizontal resolution from JAXA
(http://www.eorc.jaxa.jp/ALOS/en/aw3d30/). The AW3D DSM has been generated from ALOS and PRISM datasets aquired
between 2006 to 2011 AW3D dataset is known to have root mean square accuracies of 5m in height and geolocation
(Tadono et al. 2015). ASTER (Advanced Space borne Thermal Emission and Reflection Radiometer) GDEM V2 (Global
Digital Elevation Model Version 2) dataset at 1 second (30m) resolution, was used for representing DSM for 2000. ASTER
DSM was obtained free of charge from the NASA Earth Explorer (https://earthexplorer.usgs.gov/). Accuracy of ASTER
GDEM has been found to vary from one site to another, from 4.5m accuracy in a watershed region in Indonesia (Suwandana
et al. 2012) to 15.1m accuracy in mountainous region in Japan (Tachikawa et al. 2011). To extract built structures (or
features that do not form part of the terrain), a continuous ground terrain (known as Digital Elevation Model, DEM) needs to
be constructed which can then be differenced from DSM (Equation 4). A multidirectional processing and slope-dependent
filtering approach was used for DEM extraction and is further described in Misra et al. (2018).

nDSM = DSM –DEM (4)

The composite Landsat (Landsat 7 for 4 years: 2009 to 2012 and Landsat 8 for other four years: 2013 to 2016) was classified
in a supervised manned using Mahalanobis distance into 7 classes (including class built-up) Using the class built-up binary
mask, corresponding pixels in the nDSM raster were considered to generate built-area nDSM. The same land cover map
(belonging to the year 2014) was applied on ASTER so as to compare height differences over the time period due to
construction activities, demolition, etc. Since higher structures are often found in areas with high economic activity, built
area nDSM was resampled and analysed with VIIRS (Visible Infrared Imaging Radiometer Suite) DNB (Day-Night Band)





composite. VIIRS DNB monthly images for the year 2014 were freely obtained (https://ngdc.noaa.gov/eog/viirs/) and used to study nightlight radiance over study locations. Annual DNB image was prepared by considering median radiance of monthly composites of DNB product. It consists of light from persistent sources but the original data has not been filtered for forest fires or any other activity that may generate light from natural sources. Its spatial resolution is approximately 15

seconds (450m).

These land use maps were used for spatial distribution of Manufacturing industrial and Residential emissions into the same grid net – 1 km resolution with Transportation sector. REASv2.1 used different method to allocate their emissions. Road network was applied to distribute Traffic emission. The spatial distribution of rural, urban, and total populations to allocate country- and sub-region-based emissions from area sources to grid cells. Obviously, the level of detail required by local

emissions inventories cannot be met if the approach of REAS is applied. In this study, we applied weights for three types of roads in road network and urban morphology maps that enable the spatial allocation into 1 km grid nets. This advantageous method can benefit the compilation of other community scale EIs in future.

### 3. Results

### 3.1. Emissions from each sector

### 3.1.1. Transportation emission

Table 10 summarizes emissions for each species in HCMC from on road transportation during eight years, from 2009 to 2016. Figure 3 shows the relative contribution of vehicle types to total Transportation emission. On the whole, all 12 pollutants expressed the same gradual growing trend over 8 years. The reason is that the increase in emissions of all species was driven by the same data set of vehicle population, VKT and emission factors were assumed to be constant. However, the

mix of contributions from different vehicle types were different for each pollutant species.

Total CO emissions in 2016 were 682 Gg, increasing by 312 Gg (+98%) compared to 2009. Over 95% of CO emission from transportation was accounted by MC. In HCMC, growth rate of MC reached 180% over 8 years (from over 4 mil. to over 7.2 mil. Veh.) Although the increase rate of personal cars was higher (+130% from 2009 to 2016), MC had constantly shared around 90% of totally vehicle population during that period (Fig. 3) Furthermore, CO emission factor of MC by far the

highest one among five types of vehicle (12.592 g.km-1.vehicle-1) This is almost 6 times higher than the one of personal car and taxi.

Over this period, NOx presented a different pattern, despite the same monotonically increasing trend with CO. Total NOx in 2016 was 84.8 Gg (+157.4%) for HCMC. The majority of NOx emission was from heavy duty vehicle (HDV) – truck (50% - 61.9% during the period 2009-2016), followed by personal car (19-21%) The fact is that HDVs use diesel engines which

emit higher amount of PM and NOx (Reşitoğlu, İ.A. et al., 2015) Besides, the truck fleet in HCMC is quite old, the average age is 11.7 years, leading to high sharing ratio of out of dated engines (75% trucks used Euro 2 engines) (N.T.K.Oanh et al., 2015) As a result, NOx emission factor of truck is the highest one among five types of vehicle. Although emission factors of truck (17 g.km-1.vehicle-1) and bus (16.954 g.km-1.vehicle-1) are roughly equal, the dominated population of trucks made it the largest contributor to total NOx emission from transportation.



Total SO2 emission from on road traffic in 2016 was 6.773 Gg. The emission values more than double from 2009 to 2016 (+155.78%) Again, contribution was dominated by fleet of truck (39-48%), followed by personal cars (38-42%) Different from CO emission, MC was not an important source for SO2. This common vehicle accounts for modest proportion (7.4 - 10.5%) compared to others. This trend reflects the highest SO2 emission factor of truck which use diesel engines (1.06 g.km-1.vehicle-1).

Total emissions of PM10/PM2.5 in 2016 from Transportation sector in HCMC were 20.4/5.07 Gg (+143/157%). Showing a similar pattern with NOx, truck made the highest contributions to emission of particle matter (38.4-49.5% and 54.7-66.9% for PM10 and PM2.5, respectively) The reason is that diesel engines emitted much more fine particles than gasoline engines that are mainly found in MC (Reşitoğlu, İ.A. et al., 2015) Emission factors of HDV used in this study are 3.28 and 1.1 g.km-1.vehicle-1 for PM10 and PM2.5 respectively, that are almost 35 and 61 times higher than those ones of MC. Consequently,

although MC shared over 90% of total vehicle population, the dominated emission factors of vehicles using diesel engines made them to be the main emission sources of PM.

Emission of aerosols – BC and OC showed almost opposite tendencies. Total emissions BC and OC emissions in 2016 in HCMC were 0.222 and 0.982 Gg (+85%/+87%) respectively. MC made the most considerable contribution of these primary aerosol emissions (92% - 94%) The remaining transportation types accounted for very small shares (1%-5%) This is because

we applied emission factors of BC and OC from Updated Emission Factors of Air Pollutants from Vehicle Operations in GREET (Greenhouse gases, Regulated Emissions, and Energy use in Transportation) using MOVES (Motor Vehicle Emission Simulator) (Hao Cai et al., 2013) According to this database, MC is the most common source of BC and OC emissions (0.004 and 0.0178 g.km-1.vehicle-1)

Regarding Green house gases, total emission of CO2, CH4 and N2O in 2016 in HCMC were 21999 Gg (+103%), 6.601 Gg

(+100%) and 0.292 Gg (+ 92%), respectively. In the cases of CO2 and N2O, MC and personal car are considered as the main sources for emissions. 50-57 % of CO2 emissions were from MC and its proportion decreased by 7% over 8 years. Personal cars ranked second but its share of the total rose from 32 to 36.8% from 2009 to 2016. The similar trend was seen in N2O, although MC had larger share for this species than CO2 (74-79%) The contribution of personal car slowly grew from 18 to 22% of total N2O emission from Transportation. In terms of CH4 which is responsible as an important measure to reduce

Short-lived climate forcers, the share of MC was by far the highest (95-97%) Very small share of CH4 emission in transportation sector were from other vehicle types. These estimations are in line with other previous studies who claimed diesel engines emit less CO2 and Greenhouse Gases than similar gasoline ones (Reşitoğlu, İ.A. et al., 2015)

### 3.1.2. Manufacturing industries and Residential building emission

Table 11 presents annual emissions from fuel consumptions in two other key sectors: Manufacturing industries and Residential building. Figure 4 shows the comparison among three key sectors for each species in HCMC from 2009 to 2016. It should be noted that only Scope 1 emissions that occur within the boundary of city are considered in this sector. Generally





speaking, both of these emission sources expressed much smaller amount of emissions and slower growth paces than Transportation.

In terms of Manufacturing industries in HCMC, total SO2 emissions from this sector in HCMC increased monotonically from 1.092 to 2.36 Gg (+116%) over 8 years. However, these amounts are still modest compared to emissions from transport activities and the gap between two sectors increased along time. In 2009, emission of Manufacturing industries was less than a half of Traffic emission. Eight years later, this proportion reduced to 34.8%. Normally, the main source of sulfur dioxide in the air is industrial activity that processes materials that contain sulfur. However, the explosion of transport activities in

HCMC led to the dominated contribution of this sector to SO2 emission. SO2 is one component of greatest concerns because controlling SO2 emission may have the important co-benefit of reducing the formation of particulate sulfur pollutants, such as fine sulfate particles. NOx emission from Manufacturing industries in 2016 was 9.739 Gg, increased by 120% over 8 years. Similar to SO2, this accounted for only a very small fraction (almost a ninth) of traffic emission. This is reasonable because NOx is produced from the reaction of nitrogen and oxygen gases in the air during fuel combustion. In large cities,

the highest amount of nitrogen oxide emitted into the atmosphere as air pollution is usually from road transport. This distance is even more profound in the case of CO. In 2016, CO emission from Manufacturing industries was 4.152 Gg. Meanwhile, Transportation sector emitted 682.613 Gg, around 160 times higher than Manufacturing industries. Besides, emission of CO from Manufacturing industries showed a moderate growth rate compared to NOx and SO2, 75% over 8 years.

Regarding primary particle matter, both PM10 and PM2.5 emissions value almost double from 2009 to 2016. Total emissions of PM10/PM2.5 in 2016 were 0.6/0.33 Gg (+96%/93%) Again, these amounts of emission are relatively insignificant compared to emissions from transport activities. However, the sharing ratio among sectors changed for the case of BC. BC emission was still mainly from Road transport but the contributions of Manufacturing industries to total emission of this Short – lived climate pollutant cannot be neglected. In 2016, Industry sector in HCMC emitted 0.14 Gg (+129%) BC

into the atmosphere, which was equal as 63% of BC emission from Transportation and this proportion was tending to increase. OC emissions differed from BC. The total OC emission from Manufacturing industries in HCMC in 2016 was 0.0071 Gg (+86.8%), which was equal to only 7.2% of the emissions from transportation activities. Organic carbon have cooling effect as they are light reflecting. Meanwhile, black carbon is light absorbing. If the ratio of warming particles is higher, sources may have less cooling effect. It implied that apart from Transportation, reducing Short lived climate Forcers

cannot disregard the share of Manufacturing industries in HCMC.

Total CO2 emission in 2016 from Manufacturing industries in HCMC was 3861.89 Gg (+114.7%) CO2 emissions generally reflects the energy consumption, infrastructure build up and economic growth. The dominance of energy consumption from traffic activities was confirmed again by the gap in CO2 emission between Manufacturing industries and Transportation sector. Emission of CH4 and N2O from Manufacturing industries in 2016 were 0.1512 Gg (+116%) and 0.03 Gg (+115%),

respectively.





The fuel consumption of Residential sector in big cities mainly includes heating, cooling, lighting, water heating, and consumer products. In GHG emission inventory prepared by JICA, the energy consumed by households in HCMC included only Kerosene and Liquefied petroleum gas (LPG) Because of tropical climate in HCMC, this energy consumption is generally for Cooking/Household stoves, the heating and water heating can be excluded. This explains for quite trivial shares

from Residential sector in total emissions of each pollutant species, although the population explosion in study area over 8 years (+27.1%) For example, CO emission from Residential building in 2016 in HCMC was 0.4189 Gg, which is equivalent to a tenth of Manufacturing industries emission. Besides, our estimation implied that the evolution of Household emissions is much slower compared to other sectors. In parallel with 27% of the population growth in HCMC over 8 years is 34.5% increase in Green house gas emitted from Residential sector. Meanwhile, CO2 emission from Transportation soared by

104% during the same period. In fact, the shares inside household energy consumption are incommensurate. Particularly in tropical region, where fuel consumption is mainly used for cooking purposes, the largest contributor is often household electricity consumption which belongs to Scope 2 emissions. The following section discusses this in more depth.

### 3.2 Scope 1 and scope 2 emissions

According to GHG emission inventory prepared by JICA, 2017, electricity consumption shares the highest proportions in terms of Manufacturing industry and Residential sectors. Therefore, the pattern is quite different from Fuel consumption emission. The emission of Scope 2 considers Green house gas - CO2 only.

Gradual increase trend in CO2 emission from electricity consumption was recorded in both Industry and Residential sectors (Fig. 5 and Tab. 12) However, Manufacturing industry showed stronger growth (+88% during 8 years). Consequently, by

2012, CO2 emission from electricity consumption of Industry had been lower than Residential. But from 2013, the emission of this sector surpassed the one from household area. In 2016, the electricity consumptions from Manufacturing industries and Residential sectors in HCMC emitted 6985.29 Gg and 6691.43 Gg into the atmosphere, respectively. Besides, the dissimilarity in emissions between fuel consumption and electricity consumption were not the same for Industry and Household area. In terms of Manufacturing industry, electricity consumption emitted almost double than fuel consumption.

Meanwhile, the GHG emission from electricity consumption of households by far exceeded the fuel consumption of this sector.

The comparison of CO2 emissions (both Scope 1 and Scope 2) among three key sectors was shown in Fig. 6 Transportation still contributed the highest ratio, its emission always valued double the one of Manufacturing industry. However, the fastest growth was observed in Manufacturing industry (+114.7%), followed by Transportation sector (+104%). The lowest

emission and the slowest evolution were observed in Residential sector. Emission from this sector was equivalent to only a third of CO2 emission from Traffic in 2016. These findings implied that the mitigation of GHG in HCMC should consider Transportation as the most important source.

### 3.3 Spatial distribution



Emission maps can reveal the spatial intensities and where emission come from. The data is valuable for residences and local authorities in these areas. It helps identify areas of pollution concentration where special activities may be needed to control pollution. Also, it provides necessary input to air quality simulation models.

Fig. 7 revealed the spatial distribution of different pollutants as sum of three key sectors over study domain in 2016. It should be noted that these 1km resolution maps show only Scope 1 emissions which are the sum of Transportation emission

and emission from Fuel consumption of two other sectors, not including emission from Electricity consumption. According to Fig. 4, Transportation emission is by far dominated than two other source types, in terms of all pollutant species. It explains for the similarity among emission maps of various pollutions shown in Fig. 7. Relatively high emission densities are found in the central business districts (CBD) like Quan 1, Quan 4 and Quan 7, because of high road densities in this area. Suburban districts demonstrated much better situation, like low emission amplitudes observed in Can Gio, Cu Chi and Binh

Chanh. Emission within each kilometer squares in CBD can be higher over 1900 times than the ones in surrounding districts. According to these maps, abatement strategies of emission in HCMC should focus on CBD to improve air quality. If regional EI like REAS is applied, the 0.25∘ resolution cannot show the spatial distribution within HCMC. Our originality is the use of satellite derived urban land-use morphological maps which allow spatial disaggregation of emissions in city scale. In previous study of H.Q.Bang, 2010, the first emission maps were developed for HCMC using road network as allocation

factor for Transportation sector, population density as allocation factor for Industrial and Residential sectors. Our finding about high emission intensities in CBD is in line with this research which stated that the highest emissions were found in the city center, where has the highest density streets. Apart from this study, other works only provided the total emissions in HCMC, dismiss the spatial disaggregation. Thus, our approach is advantageous, it enables the mapping of emission with high reliability level in this city in future.


## 3.4 Comparison with other inventories

### 3.4.1. Comparison of transportation emission inventories

The transportation emission estimated in this study was compared with four previous studies about vehicle emission in HCMC (Tab. 13) Study of N.T.K.Oanh et al, 2015 applied the same method with this study. Activity data were number of

active vehicles, divided by 5 vehicle types and daily VKT of each vehicle type. Besides, this research considered the daily number of startups per vehicle categories and average speed to estimate detailed emission factors. EFs were separated into start up EFs and running EFs. Their output is annual emission in HCMC in 2013 for CO, VOC, NOx, SO2, PM, BC, OC, CO2, N2O and CH4.

The second study is GHG emission inventory of JICA. This study applied different method: activity data was fuel

consumption of Transportation sector (mainly gasoline and diesel), EFs were extract from 2006 IPCC Guidelines.

Another GHG emission inventory for HCMC was study of L.T.P.Linh, 2018. This author used activity data that were vehicle counts, by type of vehicles and daily VKT as study of N.T.K.Oanh et al, 2015. Their vehicle counts derived from field measurement and vehicle registry data. Regarding CO2 emission factor, they used country-specific EFs from COPERT





model instead of EFs of 2006 IPCC Guidelines. Noticeably, they divided vehicle fleet into 4 types only: MC, bus, diesel car
and gasoline car.

Another high detail level vehicle emission inventory in HCMC was prepared by H.Q.Bang, 2010. The activity data was
hourly traffic counts, including 5 categories, namely car, light truck, heavy truck, bus and MC. EFs were extracted from
literature review and were assumed to be constant on each street category and constant in time. The output is hourly
emission from vehicle fleet in HCMC. The contribution percentage of each vehicle type for each pollutant in this study was
compared with the estimation in this study.

Our CO2 emission in 2013 was quite close with the calculation of JICA although they applied the different approach to
estimate emission from Transportation. But our finding is higher than the estimation of N.T.K.Oanh et al, 2015, 2015 around
4000 Gg. We applied the same numbers of active vehicles and daily VKT used in research of N.T.K.Oanh et al, 2015 but
they used EFs in much more detail levels As a result, the different emission factors are likely the reason of this gap. In
compare with the finding of L.T.P.Linh, 2018, our CO2 emission in 2016 is double. As mentioned before, this author
classified vehicle fleet in HCMC into only 4 types, without considering truck. Besides, the difference in daily VKT, vehicle
population, and emission factor is likely to contribute to the inconsistency with our calculation.

In terms of other pollutants, estimations were lower by factors of 2 - 10 in compare with study of N.T.K.Oanh et al, 2015,
2015. The vehicle populations and daily VKT was the same for both of two studies. The smallest gap was observed for NOx
emission, while BC and OC emission showed significant distinctions. The emission factor dataset used in the study of
N.T.K.Oanh et al, 2015 was not clarified in their publication but it is expected to explain the gap between two researches.
Their study applied International Vehicle Emission (IVE) model to produce the emission factors that are relevant to the local
driving conditions and local fleet composition and considers the engine technology distribution in vehicle fleet. Meanwhile,
this study applied constant emission factors from different previous researches about transportation emissions in HCMC,
Hanoi and China.

Regarding the sharing ratio of emission from MC and personal car (PC) in this study and previous studies for 2010 and 2013,
our results is relatively consistent with research of H.Q.Bang (Tab. 14) MC is responsible for over 94% of CO emission from
on-road traffic, 29% and 15.6 % of NOx emissions in study of H.Q.Bang, 2010 and in this study, respectively. In compare
with the study of N.T.K.Oanh et al, 2015, the sharing proportion of MC from my estimation is higher but the contribution
from personal cars is lower in terms of CO. The significant gap was recorded in the sharing percentage of MC for NOx
emission. According to their calculation, MC fleet accounts for 80% of total NOx emission from transportation in HCMC.
This ratio was much higher than finding of H.Q. Bang (29%) for vehicle emission in 2010, also. As mentioned before, this
study applied the same number of active vehicles, the same daily VKT data with study of N.T.K.Oanh et al, 2015  for
emission in 2013. As a result, this gap is expected to come from inconsistent EF datasets between two studies.


### 3.4.2. Comparison with REAS v2.1 inventory



The general information of REASv2.1 and the data sources applied for three sectors: Transportation, Manufacturing industry and Residential sectors are mentioned in Supplementary part (Tab. S1 and S2) REAS2.1 mainly used the national statistical data for their activity data. Then the spatial allocation was based on road network or population data to create grid maps with
0.25 deg resolution. Table 15 shows the comparison between the estimations in this study for 2009 and the estimation of REAS 2.1 for three key sectors in HCMC in 2008. The transportation emissions from REAS for 2008 were much lower than our calculation for 2009, except BC, by factors of 1.5 to 10, depending on pollutant species. It is worth noticing that the data sources applied in two researches were not the same. REAS based on vehicle numbers, annual distance travelled, and emission factors to estimate vehicle emission. Their vehicle population were national one then total emission was allocated
to HCMC using road network, so it is likely that the calculation underestimated the emission from a traffic hot spot as HCMC. In addition, the gap between their annual VKT and our daily VKT could be a reason.

Conversely, the estimation of Industry and Residential emission of REASv2.1 surpassed the findings in this study by factors of 3-104. The difference for Residential sector is more significant than Industry. In this comparison, only Scope 1 part of HCMC emission was compared with REAS emissions. So both of two studies applied fuel consumption as activity data. But
REAS fuel consumption data was national data provided by International Energy Agency (IEA) Energy Balances database and the data applied in this study is annual sale data provided by HCMC Department of Industry and Trade (DOIT) and fuel companies. Apart from different sources of activity data, Table 16 compares the EFs of fuel consumption in these two sectors that were applied in two research. Besides, for countries which do not have their own emission inventories, REAS adopted emission factors from 1980 to 2003 from many sources, including Asian emission inventories. Meanwhile, this
study applied EFs provided by the Atmospheric Brown Cloud - Emission Inventory Manual (ABCEIM) by Shrestha et al.. (2013). REAS 2.1 applied EFs of oil and gas only. In terms of Industry sector, EFs of NOx and NMVOC were pretty similar. But EFs of CO and SO2 from REAS are much higher. CO emission factor was over double the one applied in this study. Regarding oil, SO2 emission factor was higher over 10 times than my EF. The trivial consistency was seen in EFs used in Residential sector as well.
The large discrepancies can be seen in sum of emissions from fuel consumptions of three key sectors also. The gap under 25% between our EI and REASv2.1 were recorded only in the cases of CO, NOx and CO2. According to the big gaps drawn from this analysis, the limitation when comparing a regional emission inventory with a local emission inventory can be implied. This inconsistency is expected due to the differences in activity data and EF databases. Again, the limitations of downscaling a regional EI into community scale can be seen here. Regional one is likely to underestimate the most profound
emission sector like Transportation and show the overestimation of other sectors when applying population as only spatial allocation index.

## 4. Summary and discussions

We developed a consistent and continuous emission inventory for three key sectors with local scale. Our objective is to fill
the gap among inherent EIs developed for HCMC before. The activity data and EFs were synthesized from various sources.



This local emission inventory includes most major air pollutants and greenhouse gases: SO2, NOx, CO, NMVOC, PM10, PM2.5, BC, OC, NH3, CH4, N2O, and CO2. The target years are from 2009 to 2016. Emissions are estimated for area within the boundary of HCMC and are allocated to grids at a 1km resolution.

In terms of Transportation, our results implied that the contribution of this sector to total emission in HCMC is the largest. Vehicle fleet in HCMC emitted over 682 Gg CO, 84.8 Gg NOx, 20.4 Gg PM10 and 22000Gg CO2 in 2016. The overall emission of this sector increased significantly from 2009 to 2016, mainly because of the explosion of vehicle population. The emissions of CO, NOx, SO2 and CO2 from traffic in 2016 in HCMC were 80%, 160%, 150% and 103% times of the ones in 2009, respectively. Among five vehicle types, MC contributed around 94% to total CO emission, 14 % to total NOx emission and 50- 60% to CO2 emission. Regarding NOx, SO2, and PM, truck is claimed as the biggest emission source and the sharing of personal cars was considerable in terms of NMVOC and CO2.

The emissions of Manufacturing industry and Residential sectors include both fuel consumption and electricity consumption. Electricity consumption is the most profound contributor. In 2016, the electricity consumption of Manufacturing industry and Residential sectors in HCMC emitted 6985 Gg and 6691 Gg of CO2, respectively, increasing by 87% and 45% in compare with 2009, respectively. Considering fuel consumption only, both these two sectors account for a very small percentage in compare with Transportation and the growing trend is slower compared to vehicle emission as well. The sum of CO2 emission from fuel consumption and electricity consumption of these two Stationary energy sectors still could not exceed Transportation sector. In 2016, vehicle fleet emitted 22000 Gg CO2, almost double Manufacturing sector. Meanwhile, Residential area contributed 7000 Gg CO2 only.

Regarding spatial allocation of three key emission sectors, the central business districts like Quan 1, Quan 4 and Quan 7 express the highest emission intensities, which can be over 1900 times of the ones in outskirt area. Thus, the policymakers must consider suitable future activities and regulations to control pollution in HCMC focusing on central regions. The estimations of this study showed the relative agreement with several local inherent EIs, in terms of total amount of emission and sharing ratio among elements of EI. However, the big gap was observed when comparing with REASv2.1. The different data sources of activity data and EFs database explained for this difference. Again, this implied the inevitable gap between regional and local EIs. This situation caused challenges to compile a consistent, continuous yet comparable data with processor EIs like REAS.

Our study applied the activity data and EFs synthesized from various sources and a number of limitations and uncertainties were noted. Regarding Transportation sector, this study assumed the constant VKT, EFs of vehicle fleet and road network over 8 years. The technology standard distribution for each vehicle type which impacts on the change in EFs was neglected as well. Apart from MC and personal car, the populations of bus, taxi and truck remains the uncertainty due to the limitation of statistical data. Because Traffic shares the highest ratio of emission among three primary sectors, if any of these factors, VKT, EFs and road network is improved, the accuracy of total emission in HCMC can be enhanced considerably.

In terms of Manufacturing and residential sectors, the activity data come from fuel consumption and electricity consumption data provided by HCMC Department of Industry and Trade (DOIT) and EVN respectively. Meanwhile, this study considers





emission within the boundary of HCMC only. The uncertainty relating to the administrative boundary of sale data provided by DOIT and EVN can impact on the accuracy of our estimations. Because industrial zones often located around ring road and around city boundary, so the including or excluding these emission zones could lead to considerable change in total emission amount. Apart from that, the grid EFs on electricity consumption were only available in three years 2013, 2014 and 2015. Electricity consumption is typically the largest emission source regarding stationary energy emission. So the limitation of these EFs could have the big impact on final GHG emission amount of HCMC. Moreover, EFs of fuel consumption and removal efficiencies for both two stationary energy sectors were assumed to be constant during 8 years, meaning the technology evolution was not considered.

The urban morphology maps applied in our study include doubt, too. We relied on only one building height data (extracted from AW3D30) in 2011 to prepare land use maps for 8 years. The assumption of constant building height neglects vertical growth and land-use transitions, causing inevitable uncertainty in spatial allocation of emission. Also, this approach assumes that all constituents of the field data of building height and land use could improve the reliability of annual urban morphology maps.

## 5.    Conclusion

With updated methods and substantial new data on local emission sources, a city-scale emission inventory of twelve air pollutants has been developed for HCMC, Vietnam for 8 years, 2009–2016. Through statistical data, emission factors adopted from previous studies in Asia and spatial emission distributions, the total emissions for three major sources (Transportation, Manufacturing industries and Residential building sectors) are estimated and mapped. Emissions in the city are dominated by Traffic activities, followed by Manufacturing industries and Household area. All these sector show the increases in emissions although the growth rates are not the same.

In future, to improve this local emission inventory, it is needed to include other sectors such as Waste, Industrial process and product use (IPPU) and Agriculture, forestry, and other land use (AFOLU) for a comprehensive EI. Besides, although HCMC is located in tropical region, the significant monthly variations in fuel consumption and electricity consumption of Stationary energy sectors is not expected, improving the level of detail of EI from annually to monthly is still required. The next step is using it as input data of atmospheric chemistry models and conduct the comparison to independently derived data. In this case, remote sensing data, observed data provided by air quality monitoring network can be the answers. In addition, with available local EIs, policy makers can see the quantitative improvement of air quality by atmospheric chemistry models using adjusted emission inventory according to mitigation solutions as input data. Besides, the improvement of local activity data and emission factors could enhance the reliability of this EI.

The continuous growth in emissions from all three sectors implies that substantial efforts should be undertaken to achieve targets in emission reduction in HCMC. According to our calculations, emission abatement should prioritize Transportation activities. To decrease the total emission of this dominant sector, a number of air pollution control solutions are proposed: exhaust control policies for MC and truck to improve their emission factors, replacement the personal vehicles with public



transport. It is a fact that the major of MC and trucks in HCMC use old technology standard engines as mentioned above, so the penetration of modern engines will lead to significant improvement of air quality in this city. In addition, because emissions from fuel consumption only account for small proportions, the reduction in grid emission factors of electricity consumption could have a remarkable impact on emissions from Manufacturing industrial and Residential building sectors. It can be achieved by the transition from coal-fired power to other forms of clean energy like hydroelectricity or solar power. According to our emission maps, the pollution control solutions should focus on Central business districts where the traffic intensities are high. This area also has the highest population density. Thus, the emission mitigations for Central business districts will benefit not only the GHG reduction, but also the improvement of human exposure to air pollution.

Our originality is the use of satellite derived urban land-use morphological maps for spatial distribution of area emission sources. Conventionally, the existing regional inventories base on surrogate statistics such as fuel consumption, employment, population as spatial proxies of grid allocation. It can introduce a large uncertainty when downscaling to community scale EI because its assumption is based on linear relationship between the proxy value and the emission. Besides, these statistics are often adopted from field-work based inventories. Although field-work based data can be highly accurate, they are labor consuming and cannot be performed frequently. The use of those existing spatial distribution surrogates neglects the effects of urban sprawl that is evident in big cities, also. It is desirable to have access to revised spatial allocation factors that may be more representative of spatial distributions in community scale and more available. And even if statistical data is inaccessible in other cities remote sensing data can be used. Remote sensing data can be updated frequently. Thus, the use of satellite images makes spatial disaggregation updating quite simple and efficient. Besides, it is the best tool to represent urban expansion and land use change, so it ensures the accuracy of grid allocation when closely related spatial activity surrogate is needed to compile EI in local scale.

**Data availability:**

Gridded emission data sets at 1km resolution for three key sectors from 2009 to 2016 are available based on request from corresponding author.

**Author contribution:**

NTQT and WT conducted the study design. NTQT contributed to actual works for development of emission inventory such as collecting data and information, settings of parameters, calculating emissions and creating final data sets. PM conducted urban morphology mapping. NTQT prepared the manuscript with contributions from WT and PM.

**Competing interest:**

The authors declare that they have no conflict of interest.





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

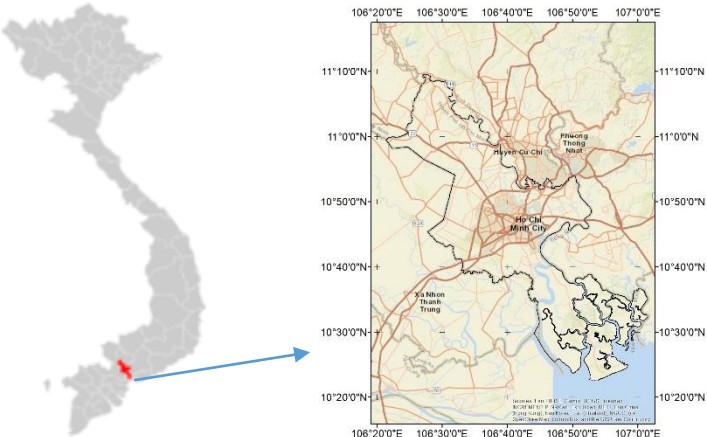

**Figure 1. Ho Chi Minh city – inventory domain of our EI (© OpenStreetMap contributors 2019. Distributed under a Creative Commons BY-SA License.)**






(a)    Transportation emission


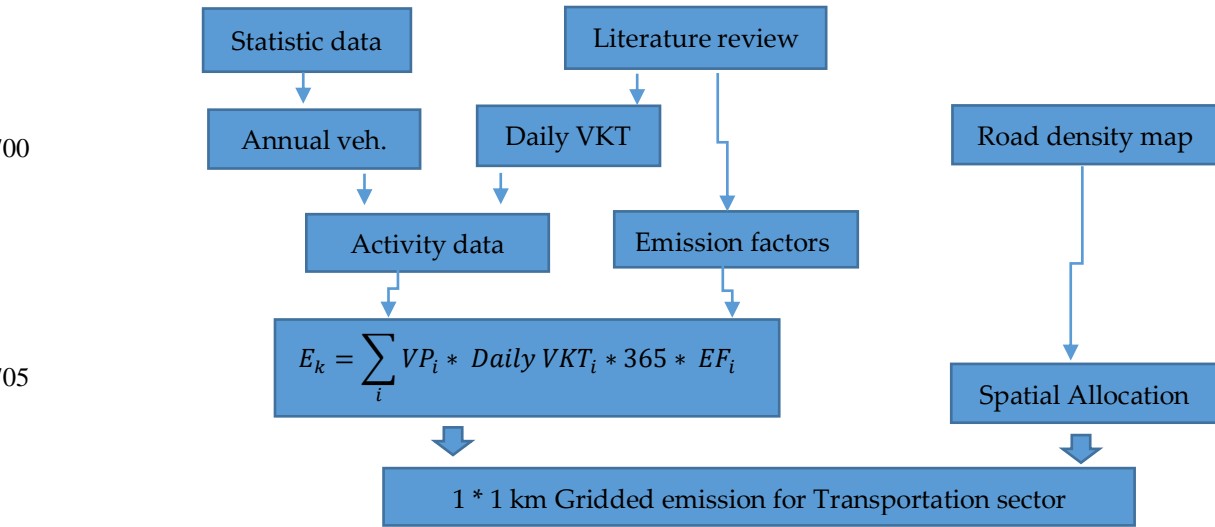


$$E_k = \sum_i VP_i * Daily\ VKT_i * 365 * EF_i$$

(b)    Manufacturing industries and Residential building emission




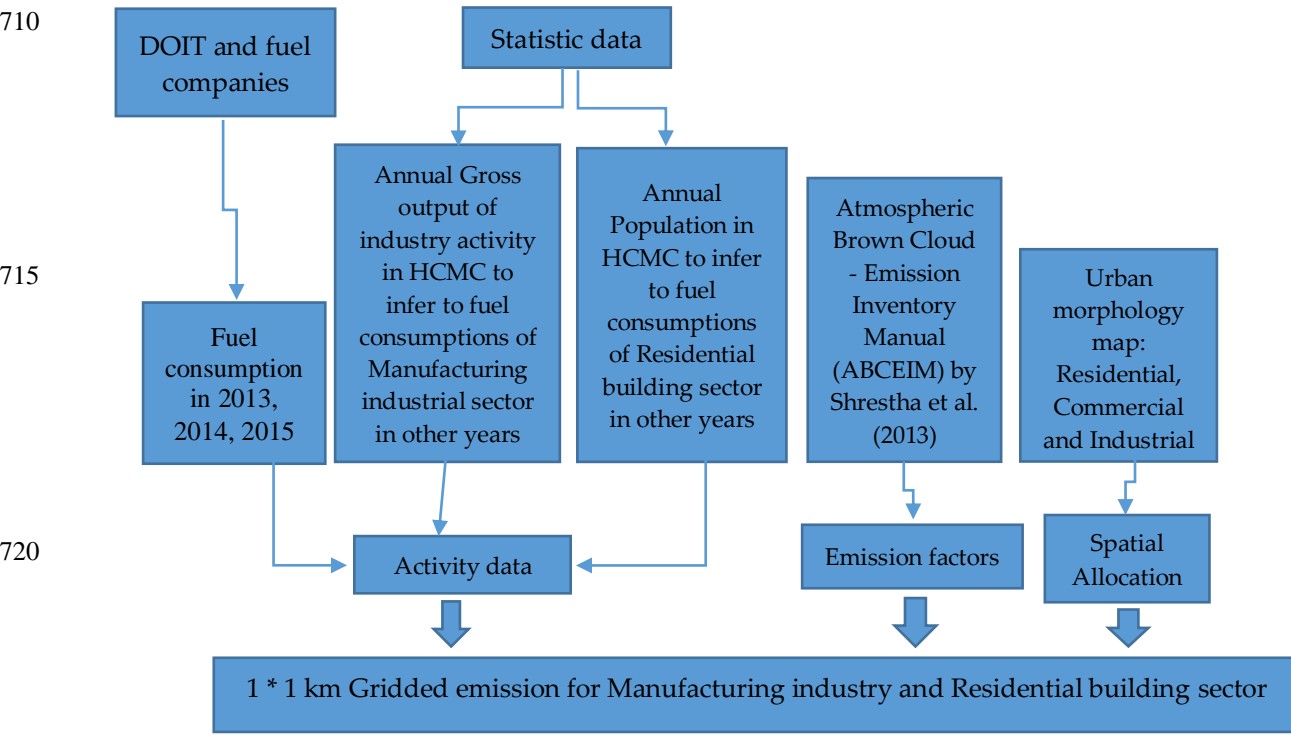

**Figure 2. Schematic flow diagrams showing estimation of emissions from (a) transportation sources, (b) Manufacturing industrial source and Residential building sources**




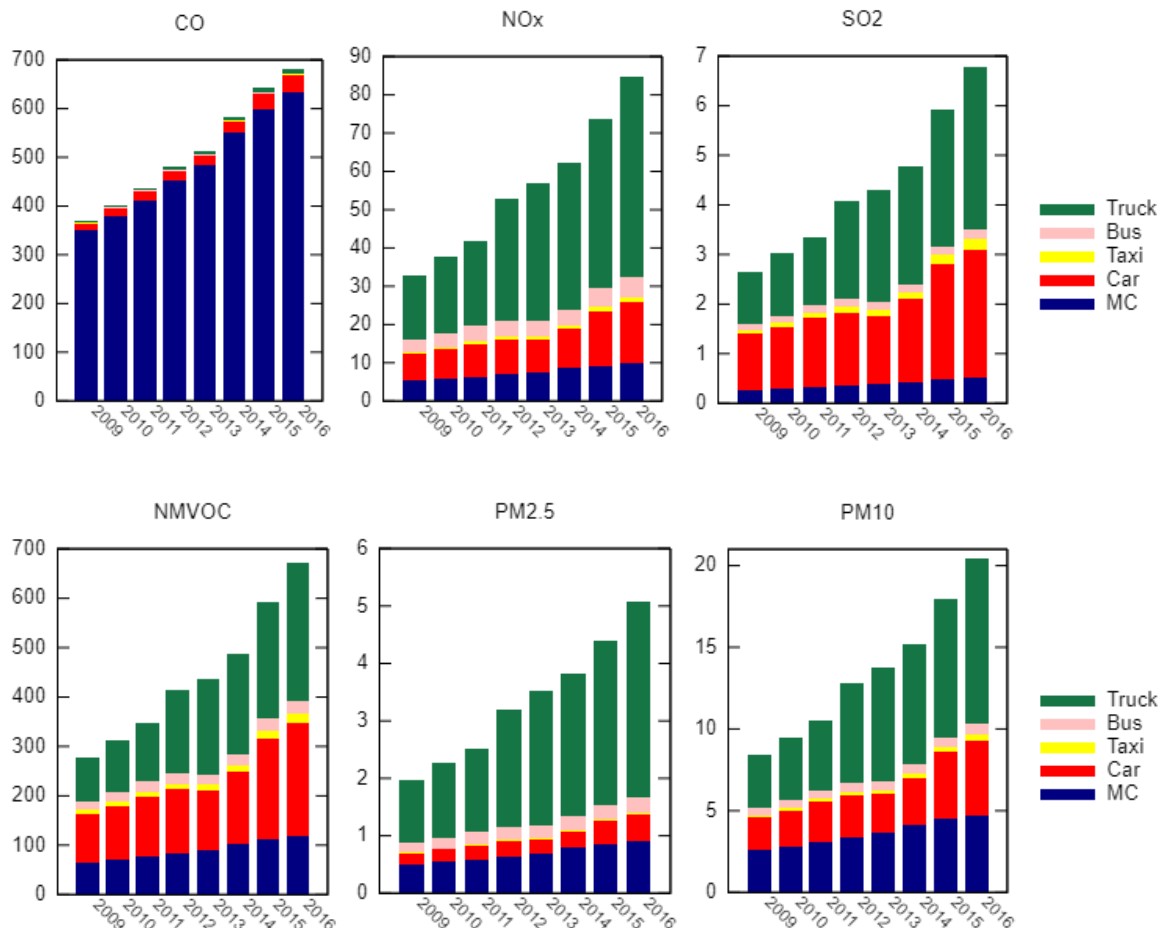





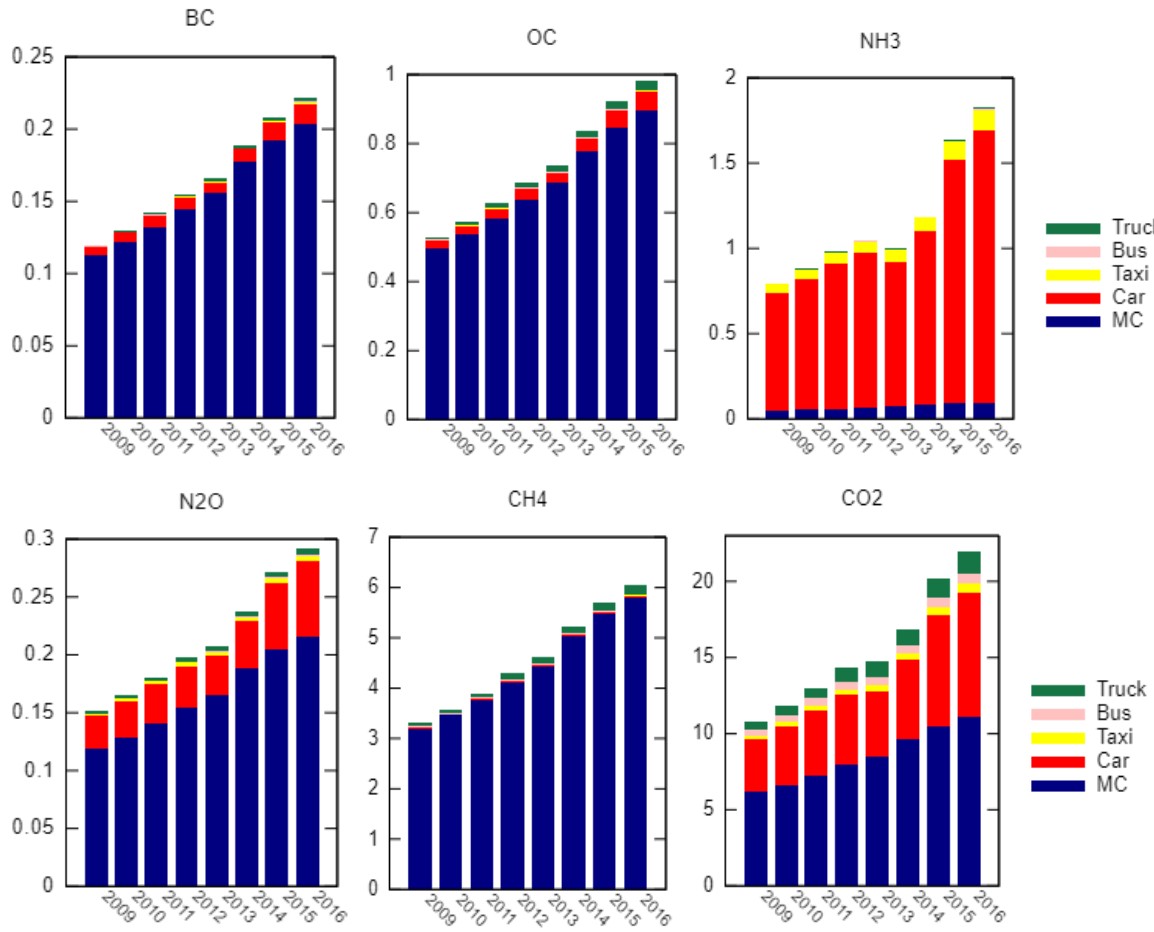


**Figure 3. Annual emissions of twelve pollutant species in HCMC from 2009 to 2016 for each vehicle type (unit: Gg)**

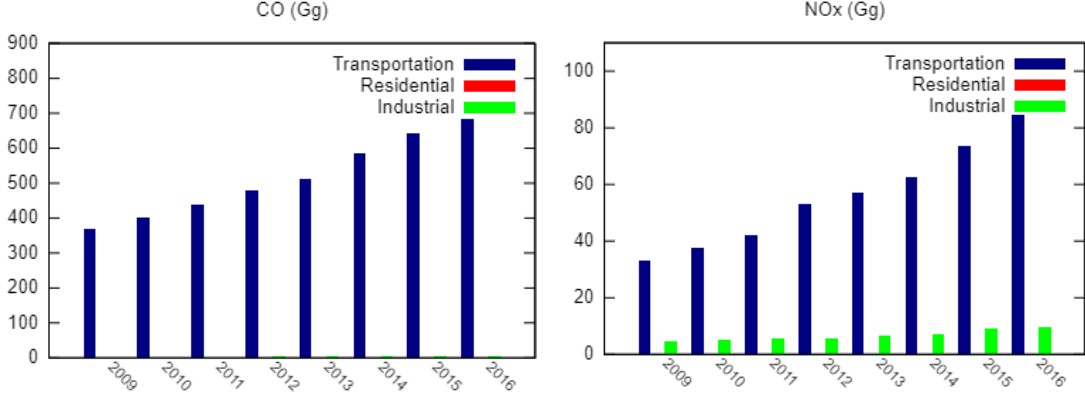





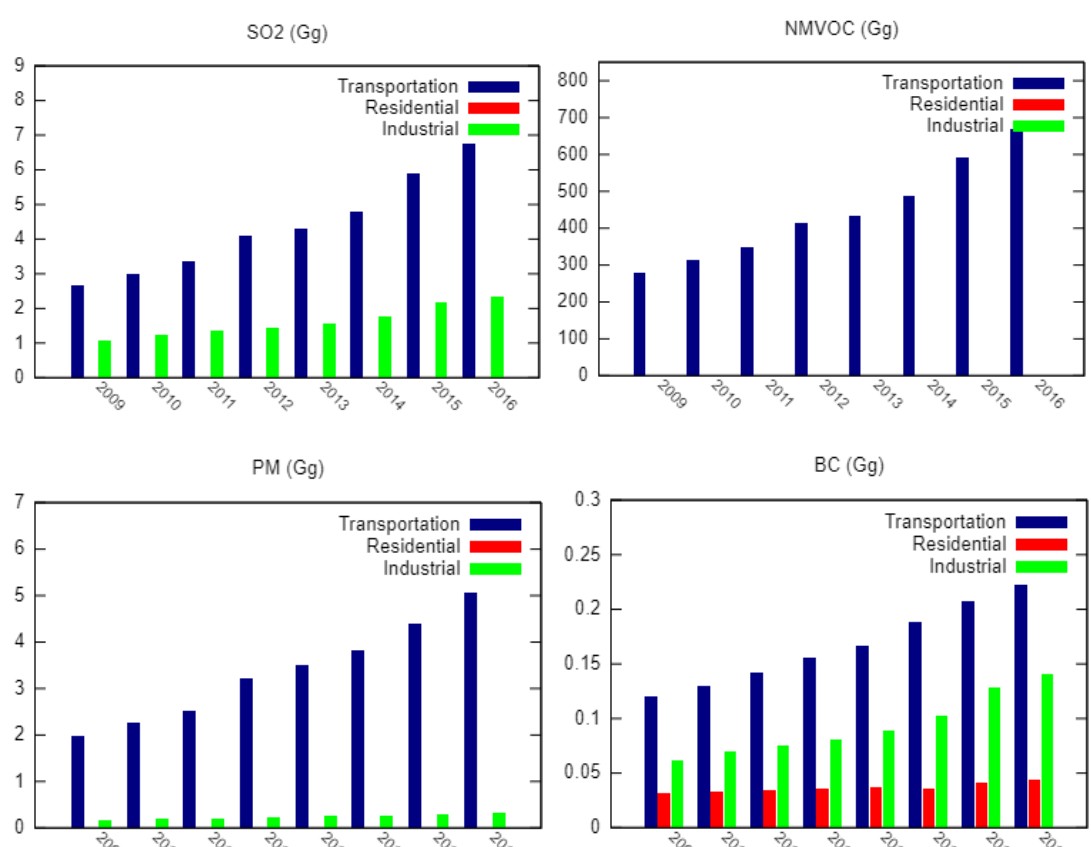






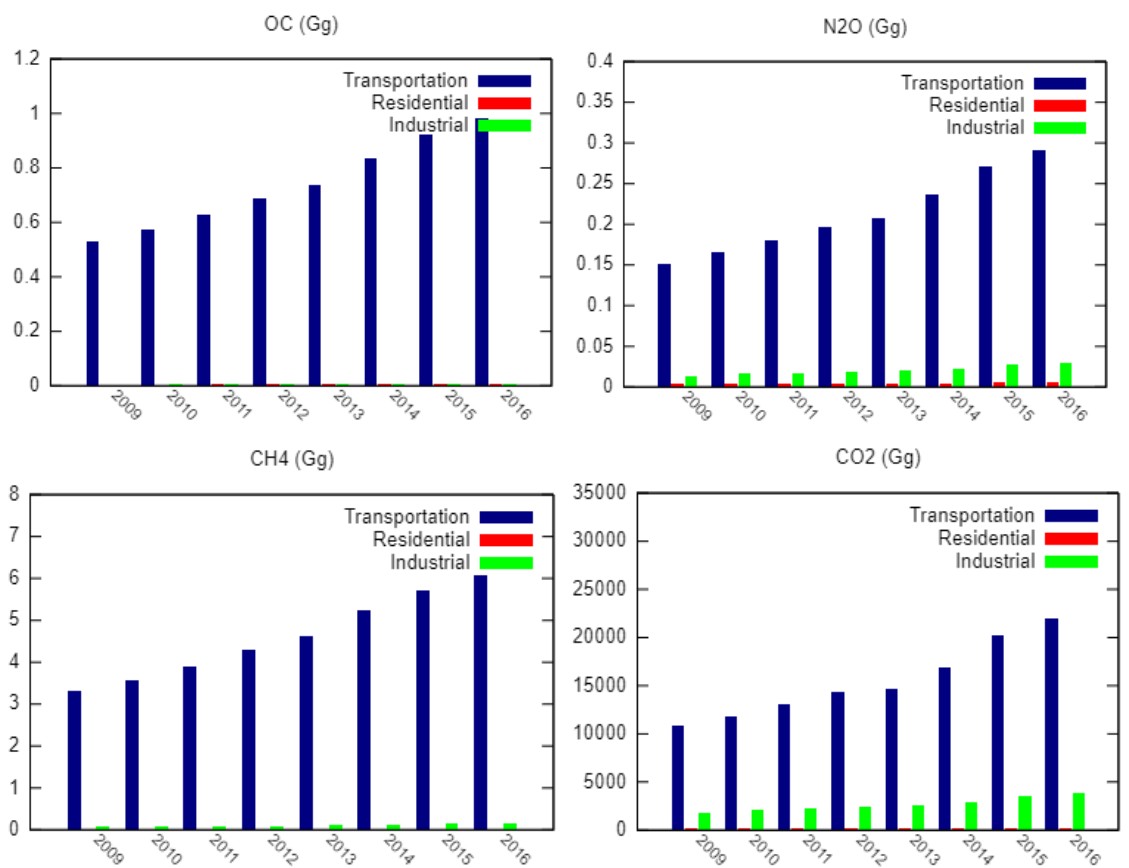

**Figure 4. Annual emissions of each species in HCMC from 2009 to 2016 for three key sectors (Unit: Gg)**

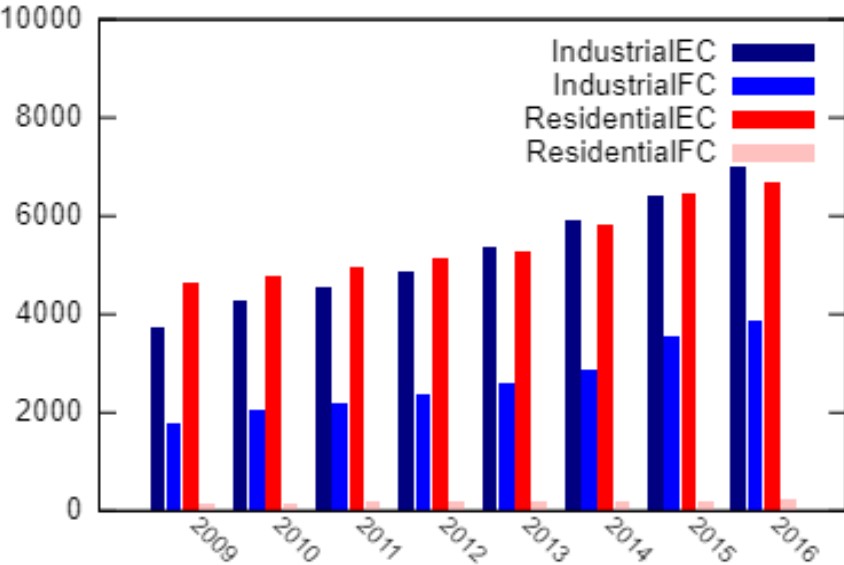





**Figure 5. Annual CO2 emissions (Unit: Gg) of Electricity consumption (EC) and Fuel consumption (FC) of Manufacturing industry and Residential sector in HCMC from 2009 to 2016.**

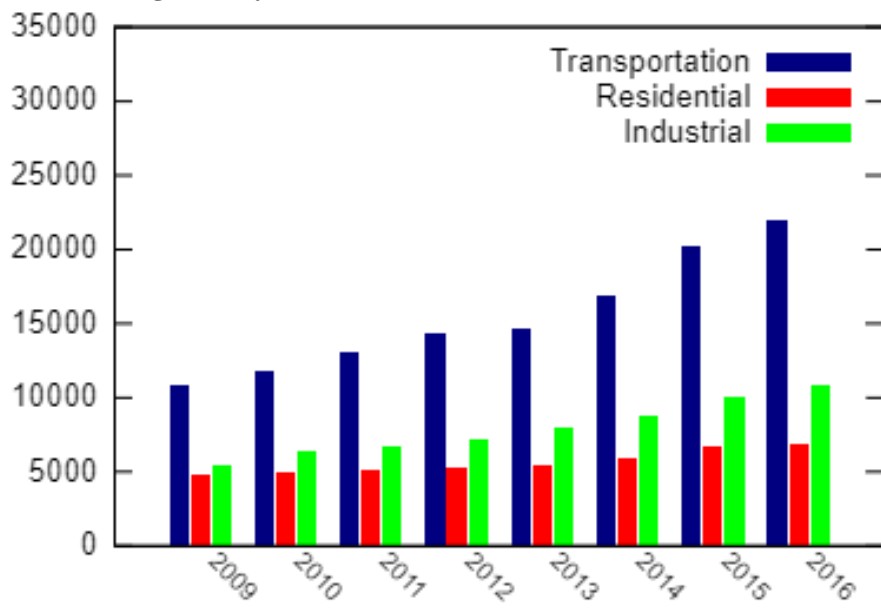

**Figure 6. Annual CO2 emissions (Unit: Gg) of three key sectors: Transportation, Manufacturing industry and Residential sector in HCMC**




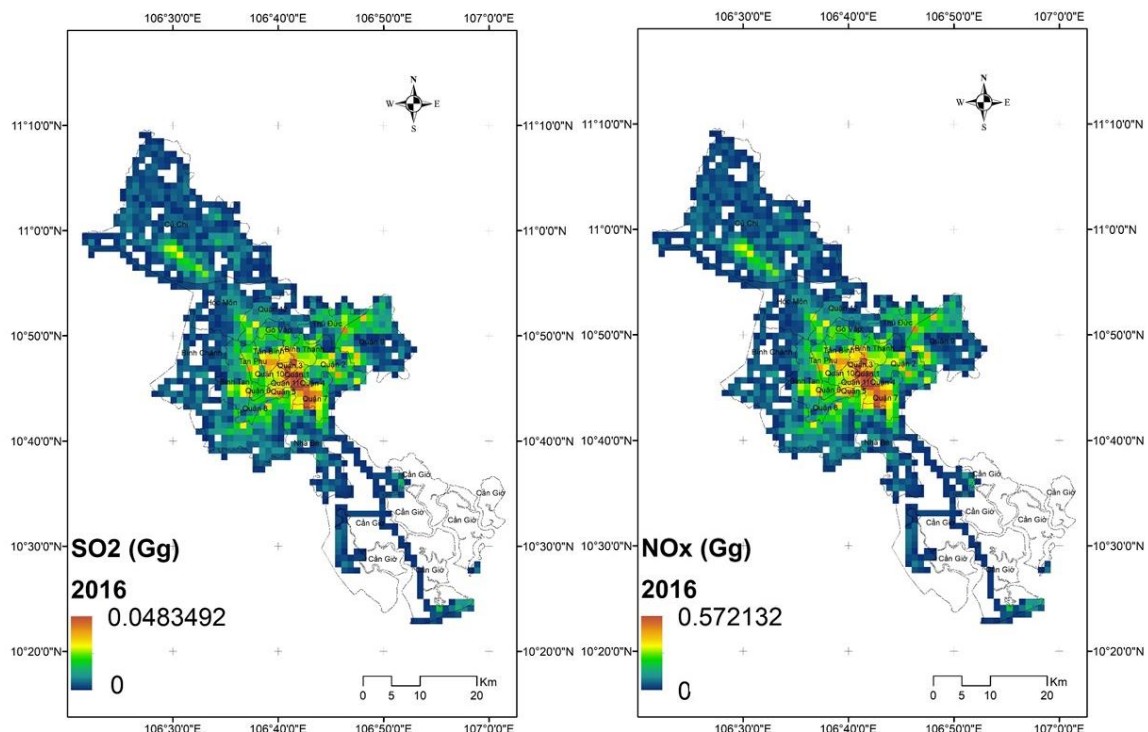



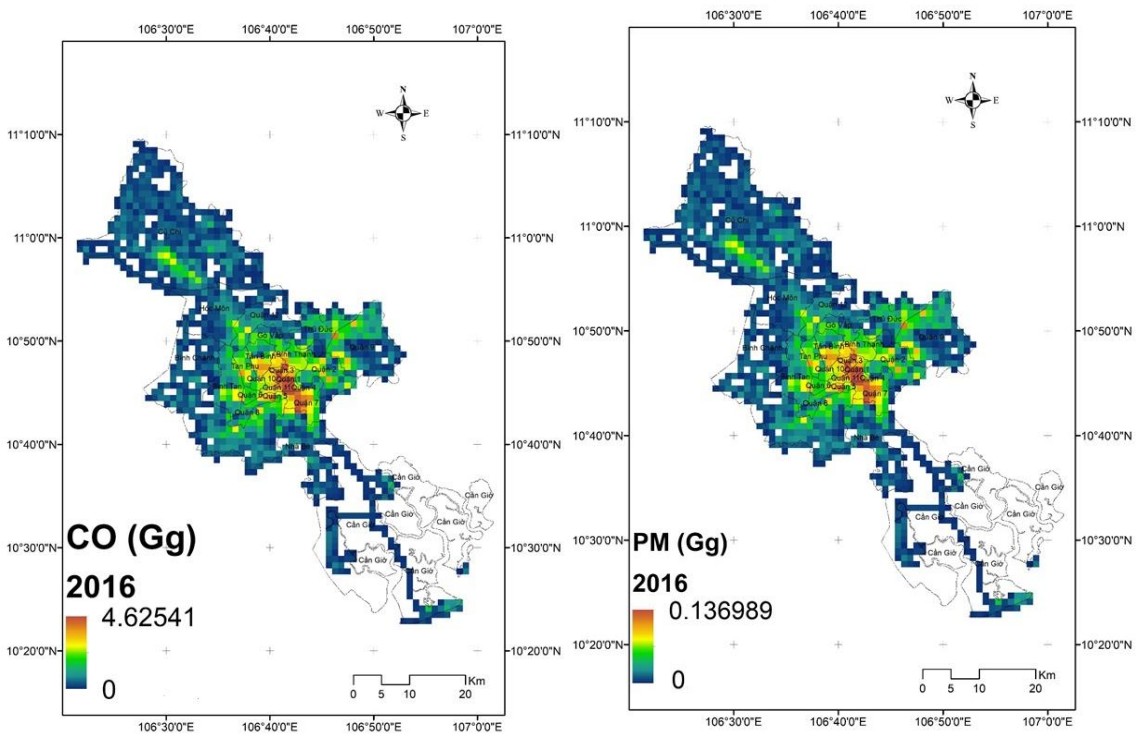

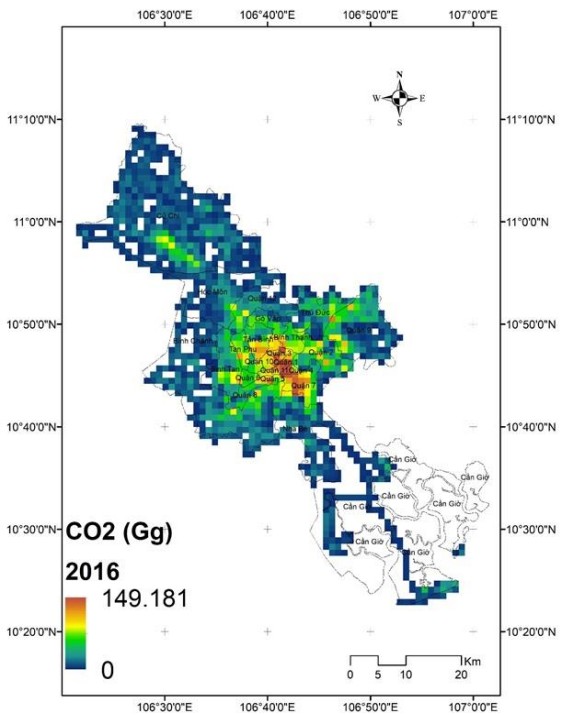



**Figure 7. Emission maps of NOx, SO2, CO, PM and CO2 in 2016 in HCMC as sum of three key sectors: Transportation, Manufacturing industry and Residential Sectors.**

**Table 1. General information on Asia emission inventories**

| Emission inventories | References | Species | Years | Area covered | Spatial resolution |
|---|---|---|---|---|---|
| | Kato and Akimoto (1992) | SO2 and NOx | 1975, 1980, 1985, 1986 and 1987 | East Asian, Southeast Asian and South Asian countries | 1∘×1∘ |
| TRACE-P | Jacob et al., 2003 | | 2000 | Over western Pacific | |
| INTEX-B | Zhang et al., 2009 | | 2006 | Over western Pacific | |
| REASv1.1 | T. Ohara et al, 2007 | SO2, NOx, CO, NMVOC, BC, OC, CO2, NH3, CH4 and N2O | From 1980 to 2020 | East, Southeast and South Asia | 0.5∘×0.5∘ |
| REASv2.1 | J. Kurokawa et al, 2013 | SO2, NOx, CO, NMVOC, PM2.5, PM10, BC, OC, CO2, NH3, CH4 and N2O | From 2000 to 2008 | East, Southeast, South Asia, Central Asia and Russia Asia | 0.25∘×0.25∘ |
| REASv3.1 | J. Kurokawa et al, 2019 | SO2, NOx, CO, NMVOC, PM2.5, PM10, BC, OC, CO2, NH3, CH4 and N2O | During 1950-1955 and from 2010-2015 | East, Southeast, South Asia, Central Asia and Russia Asia | 0.25∘×0.25∘ |
| MIX | Tsinghua University (Zhang et al., 2009; Li et al., 2014; Zheng et al., 2014) | SO2, NOx, CO, NMVOC, NH3, PM10, PM2.5, BC, OC and CO2 | 2008 and 2010 | East, Southeast, South Asia, Central Asia and Russia Asia | 0.25∘×0.25∘ |


**Table 2. General information on HCMC emission inventory**

| Item | Description for targets |
|---|---|
| **Species** | SO2, NOx, CO, NMVOC, BC, OC, CO2, NH3, CH4, N2O, PM10 and PM2.5 |
| **Years** | 2009 - 2016 |
| **Area** | Ho Chi Minh city, Vietnam |
| **Emission sectors** | (1) Transportation, (2) Manufacturing industries and (3) Residential building |
| **Spatial resolution** | 1 km |





| Temporal resolution | Annually |
|---|---|

**Table 3. Average daily vehicle kilometre travelled of vehicle types in HCMC (N.K. Oanh and H.H. Van, 2015)**

| Vehicle types | Average daily vehicle mileage traveled (km/day) |
|---|---|
| Motorcycle | 19 |
| Bus | 195.6 |
| Taxi | 124 |
| Personal car | 33.4 |
| Truck | 31.4 |

**Table 4. Number of registered vehicles by type in HCMC over years**

| | MC | Car | Taxi | Bus | Truck |
|---|---|---|---|---|---|
| 2009 | 4013208 [b] | 257132 [b] | 10300 [c] | 2814 [d] | 85623 [e] |
| 2010 | 4340530 [b] | 283810 [b] | 12600 [c] | 3016 [d] | 101961 [e] |
| 2011 | 4721123 [b] | 317816 [b] | 13900 [c] | 3370 [d] | 114052 [e] |
| 2012 | 5171000 [b] | 337743 [b] | 15000 [c] | 3587 [d] | 162676 [e] |
| 2013 | 5558000 [a] | 315943 [a] | 15500 [a] | 3358 [a] | 185501 [a] |
| 2014 | 6318000 [b] | 379763 [b] | 17000 [c] | 3596 [d] | 197057 [e] |
| 2015 | 6863707 [b] | 532835 [f] | 23853 [d] | 3833 [d] | 226677 [e] |
| 2016 | 7266000 [b] | 595349 [f] | 26651 [d] | 4283 [d] | 269294 [e] |

(a)    N.T.K.Oanh et al, 2015

(b)    Statistical data provided by The Transport Department of HCMC.

(c)    JICA, Report on Ho Chi Minh City – Osaka City Cooperation Project for Developing Low Carbon City, 2016.

(d)    Proportional estimation basing on number of cars.

(e)    Proportional estimation basing on annual volume of freight carried that were provided by HCMC Statistical Yearbook.

(f)    L.P.Linh et al, 2018

**Table 5 The emission factors (g.km-1.vehicle-1) from literature review**

| Pollutant | MC | Car and Taxi | Bus | Truck |
|---|---|---|---|---|
| CO | 12.592[b] | 2.21 [b] | 6.905 [a] | 3.1 [b] |
| NOx | 0.195 [b] | 1.05 [b] | 16.954 [a] | 17 [b] |
| SO2 | 0.01 [c] | 0.17 [b] | 0.64 [b] | 1.06 [b] |
| CH4 | 0.115 [d] | 0.0031 [d] | 0.077 [d] | 0.062 [d] |
| PM2.5 | 0.018 [f] | 0.03 [f] | 0.9 [f] | 1.1 [f] |
| PM10 | 0.094 [c] | 0.3 [b] | 2.08 [a] | 3.28 [b] |
| NMVOC | 2.34 [e] | 15.02 [e] | 89.92 [e] | 89.92 [e] |





| | | | | |
|---|---|---|---|---|
| **BC** | 0.004039 [f] | 0.000932 [f] | 0.00112 [f] | 0.000746 [f] |
| **OC** | 0.0178 [f] | 0.00342 [f] | 0.012 [f] | 0.00808 [f] |
| **NH3** | 0.0019 [g] | 0.1043 [g] | 0.0029 [g] | 0.0029 [g] |
| **N2O** | 0.00429 [f] | 0.00423 [f] | 0.0018 [f] | 0.001926 [f] |
| **CO2** | 221 [g] | 530 [g] | 2050 [g] | 486 [g] |


Source: (a) T.T.Trang et al, 2015 (Study in Hanoi)

(b) N.T.Hung et al, 2014 (Study in Hanoi)

(c ) N.T.Kim Oanh et al, 2012 (Study in Hanoi)

(d) Rui-Qiang Yuan et al, 2016 (Study in China)

(e ) Belalcazar et al., 2009; Ho et al., 2008 (Studies in HCMC)

(f) Hao Cai et al, 2015 (Updated Emission Factors of Air Pollutants from Vehicle Operations in GREETTM Using MOVES)

(g) EMEP/EEA air pollutant emission inventory guidebook 2016, updated in 2018.

**Table 6 Annual fuel consumption in HCMC in 2013, 2014, 2015 and Ratio of Final Fuel Consumption by Sub-Sector**
**(Manufacturing industrial and Residential sectors) and Fuel Type in Vietnam in 2014 provided by JICA, 2017**

| | | Fuel consumption (TJ/ year) | | | Ratio of Final Fuel Consumption by Sub-Sector in Vietnam in 2014 (%) | |
|---|---|---|---|---|---|---|
| | **Fuel type** | **2013** | **2014** | **2015** | **Manufacturing industrial sector** | **Residential sector** |
| 1 | Gasoline | 115855 | 119247 | 134544 | 0 | 0 |
| 2 | Diesel | 120218 | 141229 | 180686 | 16% | 1% |
| 3 | Heavy oil | 15976 | 16540 | 19334 | 86% | 1% |
| 4 | Kerosene | 1664 | 1607 | 1901 | 12% | 74% |
| 5 | LPG | 2268 | 2246 | 2541 | 15% | 55% |
| 6 | Natural gas | 1463 | 1441 | 1567 | 100% | 0% |

**Table 7 Emission factors for Manufacturing industrial and construction (a) and Residential (b) sectors from the compiled database provided by the Atmospheric Brown Cloud - Emission Inventory Manual (ABCEIM) by Shrestha et al. (2013) PM2.5 and PM10 was merged into PM for Residential sector in this database (except SO2) (unit: kg/ TJ)**

| unit: kg/ TJ | Diesel | | Heavy oil | | Kerosene | | LPG | | Natural gas | |
|---|---|---|---|---|---|---|---|---|---|---|
| | **(a)** | **(b)** | **(a)** | **(b)** | **(a)** | **(b)** | **(a)** | **(b)** | **(a)** | **(b)** |
| CO | 15.00 | - | 15.00 | - | 15.00 | 167.57 | 10.00 | 78.65 | 2000.00 | - |
| NOx | 222.00 | - | 145.00 | - | 167.00 | 24.94 | 56.00 | 37.21 | 53.00 | - |
| CH4 | 3.00 | - | 3.00 | - | 3.00 | 2.04 | 1.00 | 2.96 | 1.00 | - |
| PM2.5 | 0.83 | - | 17.00 | - | 10.00 | - | - | - | 0.04 | - |
| PM10 | 3.30 | - | 27.40 | - | 10.80 | 43.08 | - | 5.50 | 0.04 | - |
| NMVOC | 5.00 | - | 5.00 | - | 5.00 | 8.84 | 5.00 | 33.83 | 5.00 | - |
| BC | 3.90 | - | 0.90 | - | 5.50 | 20.41 | - | 4.23 | 0.00 | - |





| | | | | | | | | | | |
|---|---|---|---|---|---|---|---|---|---|---|
| OC | 0.00 | - | 0.37 | - | 1.70 | 2.04 | - | 1.06 | 0.02 | - |
| NH3 | 0.01 | - | 0.10 | - | - | - | - | - | 1.31 | - |
| N2O | 0.60 | - | 0.60 | - | 0.60 | 1.59 | 0.10 | 1.90 | 0.10 | - |
| CO2 | 74100.00 | - | 77400.00 | - | 71900.00 | 70975.06 | 63100.00 | 63002.11 | 56100.00 | - |

- Not available.

**Table 8. Annual Gross output of industry at current prices by industry activity in HCMC and Population of HCMC over years provided by HCMC Statistical Yearbook**

| Year | 2009 | 2010 | 2011 | 2012 | 2013 | 2014 | 2015 | 2016 |
|---|---|---|---|---|---|---|---|---|
| Annual Gross output of industry at current prices by industry activity (Mil. USD) | 22.43 | 25.74 | 27.61 | 29.48 | 32.51 | 35.5 | 38.1 | 41.36 |
| Population (1000 people) | 5981 | 6189 | 6406 | 6629 | 6861 | 7100 | 7348 | 7605 |

**Table 9.** Electricity consumption of Manufacturing Industries and Construction and Residential sectors and grid emission factors in HCMC in 2013, 2014 and 2015, provided by Electricity of Vietnam (EVN)

| Item | 2013 | 2014 | 2015 |
|---|---|---|---|
| Electricity consumption from Manufacturing Industries and Construction sector (kWh/year) | 7186161.416 | 7557369.663 | 8094021.380 |
| Electricity consumption from Residential sector (kWh/year) | 7073622.593 | 7452131.412 | 8132452.777 |
| Grid emission factors (ton of $CO_2$/MWh) | 0.7495 | 0.7802 | 0.7950 |

**Table 10. Annual emissions for each species from Transportation sector in HCMC from 2009 to 2016 (Gg yr-1)**

| Unit (Gg) | 2009 | 2010 | 2011 | 2012 | 2013 | 2014 | 2015 | 2016 |
|---|---|---|---|---|---|---|---|---|
| CO | 370.496 | 401.503 | 437.401 | 479.761 | 513.068 | 583.732 | 641.922 | 682.613 |
| NOx | 32.933 | 37.632 | 41.908 | 52.842 | 56.972 | 62.330 | 73.576 | 84.785 |
| SO2 | 2.648 | 3.012 | 3.360 | 4.087 | 4.290 | 4.784 | 5.913 | 6.773 |
| CH4 | 3.299 | 3.575 | 3.892 | 4.288 | 4.610 | 5.231 | 5.702 | 6.061 |
| PM2.5 | 1.973 | 2.257 | 2.507 | 3.207 | 3.512 | 3.819 | 4.403 | 5.072 |
| PM10 | 8.372 | 9.466 | 10.501 | 12.822 | 13.738 | 15.215 | 17.990 | 20.441 |
| NMVOC | 277.534 | 312.825 | 347.725 | 414.957 | 435.228 | 486.614 | 591.171 | 670.503 |
| BC | 0.120 | 0.130 | 0.142 | 0.155 | 0.166 | 0.189 | 0.208 | 0.222 |





| | | | | | | | | |
|---|---|---|---|---|---|---|---|---|
| OC | 0.530 | 0.575 | 0.626 | 0.688 | 0.736 | 0.837 | 0.922 | 0.982 |
| NH3 | 0.793 | 0.880 | 0.983 | 1.049 | 0.999 | 1.187 | 1.637 | 1.825 |
| N2O | 0.152 | 0.165 | 0.181 | 0.197 | 0.207 | 0.237 | 0.272 | 0.292 |
| CO2 | 10784 | 11824 | 13020 | 14309 | 14712 | 16879 | 20162 | 21999 |

**Table 11. Annual emissions from fuel consumptions in Manufacturing industrial and construction (a) and Residential building (b) sector (PM2.5 and PM10 are merged into PM in Residential sector according to ABCEIM by Shrestha et al, 2013)**

| Unit (Gg) | | CO | NOx | SO2 | CH4 | PM2.5 | PM10 | NMVOC | BC | OC | NH3 | N2O | CO2 |
|---|---|---|---|---|---|---|---|---|---|---|---|---|---|
| 2009 | (a) | 2.365 | 4.411 | 1.092 | 0.070 | 0.174 | 0.305 | 0.121 | 0.061 | 0.004 | 0.002 | 0.014 | 1798.585 |
| | (b) | 0.311 | 0.074 | 0.008 | 0.004 | | 0.010 | 0.049 | 0.032 | 0.004 | 0.000 | 0.004 | 163.829 |
| 2010 | (a) | 2.713 | 5.061 | 1.253 | 0.080 | 0.199 | 0.350 | 0.138 | 0.070 | 0.004 | 0.003 | 0.016 | 2063.558 |
| | (b) | 0.321 | 0.077 | 0.009 | 0.004 | | 0.010 | 0.050 | 0.033 | 0.004 | 0.000 | 0.004 | 169.527 |
| 2011 | (a) | 2.910 | 5.428 | 1.344 | 0.086 | 0.214 | 0.376 | 0.149 | 0.075 | 0.005 | 0.003 | 0.017 | 2213.531 |
| | (b) | 0.333 | 0.079 | 0.009 | 0.004 | | 0.011 | 0.052 | 0.034 | 0.004 | 0.000 | 0.005 | 175.471 |
| 2012 | (a) | 3.107 | 5.796 | 1.435 | 0.092 | 0.228 | 0.401 | 0.159 | 0.080 | 0.005 | 0.003 | 0.018 | 2363.503 |
| | (b) | 0.344 | 0.082 | 0.009 | 0.004 | | 0.011 | 0.054 | 0.036 | 0.004 | 0.000 | 0.005 | 181.579 |
| 2013 | (a) | 3.427 | 6.392 | 1.582 | 0.101 | 0.252 | 0.442 | 0.175 | 0.089 | 0.006 | 0.003 | 0.020 | 2606.629 |
| | (b) | 0.356 | 0.085 | 0.010 | 0.005 | | 0.011 | 0.056 | 0.037 | 0.005 | 0.000 | 0.005 | 187.934 |
| 2014 | (a) | 3.441 | 7.206 | 1.761 | 0.113 | 0.263 | 0.467 | 0.194 | 0.102 | 0.006 | 0.004 | 0.022 | 2891.343 |
| | (b) | 0.346 | 0.083 | 0.010 | 0.005 | | 0.011 | 0.055 | 0.036 | 0.004 | 0.000 | 0.005 | 183.390 |
| 2015 | (a) | 3.824 | 8.971 | 2.174 | 0.139 | 0.309 | 0.554 | 0.239 | 0.129 | 0.007 | 0.004 | 0.028 | 3557.523 |
| | (b) | 0.405 | 0.096 | 0.011 | 0.005 | | 0.013 | 0.063 | 0.042 | 0.005 | 0.000 | 0.006 | 212.917 |
| 2016 | (a) | 4.152 | 9.739 | 2.360 | 0.151 | 0.335 | 0.601 | 0.259 | 0.140 | 0.007 | 0.004 | 0.030 | 3861.896 |
| | (b) | 0.419 | 0.099 | 0.011 | 0.005 | | 0.013 | 0.065 | 0.043 | 0.005 | 0.000 | 0.006 | 220.364 |

**Table 12. Annual CO2 emissions from electricity consumptions in Manufacturing industrial and construction sector and Residential sector (emission for years marked with * were calculated from electricity consumptions provided by JICA, 2017, while other emissions for other years were calculated proportionally with Annual Gross output of industry at current prices by industry activity and Annual population in HCMC)**

| Unit (Gg) | 2009 | 2010 | 2011 | 2012 | 2013* | 2014* | 2015* | 2016* |
|---|---|---|---|---|---|---|---|---|
| Manufacturing industrial and construction | 3716.38 | 4263.89 | 4573.78 | 4883.66 | 5386.03 | 5896.26 | 6434.75 | 6985.29 |
| Residential | 4621.68 | 4782.41 | 4950.09 | 5122.41 | 5301.68 | 5814.15 | 6465.30 | 6691.43 |






**Table 13. Comparison of transportation emission estimated in this study with emission calculated in previous studies for 2013 and 2016.**

| Unit (Gg) | Van. H. H et al, 2015 2013 | JICA, 2017 2013 | L.T.P.Linh, 2018 2016 | This study 2013 | This study 2016 |
|---|---|---|---|---|---|
| CO | *1252* | | | *513.068* | |
| NOx | *61* | | | *56.972* | |
| CH4 | *33* | | | *4.6102* | |
| BC | *1.77* | | | *0.166* | |
| OC | *6.65* | | | *0.736* | |
| N2O | *0.5* | | | *0.207* | |
| CO2 | *10722* | 14693 | 10890 | *14711.59* | 21998.72 |

**Table 14. Comparison of sharing ratios of emission from MC and personal car (PC) in this study and previous studies for 2010 and 2013 (Unit: %)**

| Unit (%) | H.Q.Bang, 2010 2010 MC | N.T.K.Oanh, 2015 2013 MC | N.T.K.Oanh, 2015 2013 PC | *This study 2010 MC* | *This study 2013 MC* | *This study 2013 PC* |
|---|---|---|---|---|---|---|
| CO | 94 | 85 | 12 | *94.4* | *94.6* | *3.5* |
| NOx | 29 | 80 | 14 | *15.6* | *13.2* | *14.9* |

**Table 15 Comparison of Transportation, Industry and Domestic emissions estimated for 2009 in this study and emissions estimated by REAS 2.1 for 2008**

| Unit: Gg | Transportation Emission in 2009 – this study | Transportation Emission in 2008 – REAS 2.1 | Industry Emission in 2009 – this study | Industry Emission in 2008 – REAS 2.1 | Residential Emission in 2009 – this study | Residential Emission in 2008 – REAS 2.1 | Sum of three sectors This study (2009) | Sum of three sectors REAS (2008) |
|---|---|---|---|---|---|---|---|---|
| CO | 370.5 | *88.05* | 2.36 | *9.1* | 0.31 | *456.85* | 373.17 | *554* |
| NOx | 32.93 | *6.81* | 4.41 | *13.19* | 0.07 | *7.73* | 37.41 | *27.73* |
| SO2 | 2.65 | *1.64* | 1.09 | *32.42* | 0.01 | *11.18* | 3.75 | *45.24* |
| CH4 | 3.3 | *0.33* | 0.07 | *2.1* | 0 | *18.02* | 3.37 | *20.45* |
| PM2.5 | 1.97 | *0.35* | 0.17 | *18.61* | 0.01 | *25.99* | 2.15 | *44.95* |
| PM10 | 8.37 | *0.36* | 0.31 | *32.26* | | | 8.68 | *32.62* |



| | | | | | | | |
|---|---|---|---|---|---|---|---|
| NMVOC | 277.53 | *24.36* | 0.12 | *1.78* | 0.05 | *70.02* | 277.7 | *96.16* |
| BC | 0.12 | *0.15* | 0.06 | *0.94* | 0.03 | *5.19* | 0.21 | *6.28* |
| OC | 0.53 | *0.1* | 0 | *2.24* | 0 | *20.36* | 0.53 | *22.7* |
| NH3 | 0.79 | *0.07* | 0 | *0.76* | 0 | *5.91* | 0.79 | *6.74* |
| N2O | 0.15 | *0.07* | 0.01 | *0.15* | 0 | *0.3* | 0.16 | *0.52* |
| CO2 | 10784 | *1414.82* | 1798.59 | *7352.87* | 163.83 | *8054.68* | 12746.42 | *16822.37* |

**Table 16 Comparison of emission factors used for Industry sector in this study and in REAS v2.1.**

| (unit: kg/ TJ) | Diesel | Heavy oil | Kerosene | *Oil (REAS)* | LPG | Natural gas | *Gas (REAS)* |
|---|---|---|---|---|---|---|---|
| CO | 15.00 | 15.00 | 15.00 | *35.30* | 10.00 | 2000.00 | *24.00* |
| NOx | 222.00 | 145.00 | 167.00 | *157.00* | 56.00 | 53.00 | *56.40* |
| SO2 | 46.20 | 49.80 | 44.60 | *538.00* | 0.20 | 0.19 | *0.24* |
| CH4 | 3.00 | 3.00 | 3.00 | - | 1.00 | 1.00 | - |
| PM2.5 | 0.83 | 17.00 | 10.00 | *6.53* | - | 0.04 | *0.00* |
| PM10 | 3.30 | 27.40 | 10.80 | *10.40* | - | 0.04 | *0.00* |
| NMVOC | 5.00 | 5.00 | 5.00 | *4.38* | 5.00 | 5.00 | *5.00* |
| BC | 3.90 | 0.90 | 5.50 | *0.48* | - | 0.00 | *0* |
| OC | 0.00 | 0.37 | 1.70 | *0.18* | - | 0.02 | *0* |
| NH3 | 0.01 | 0.10 | - | - | - | 1.31 | - |
| N2O | 0.60 | 0.60 | 0.60 | - | 0.10 | 0.10 | - |
| CO2 | 74100.00 | 77400.00 | 71900.00 | - | 63100.00 | 56100.00 | - |

**Table 17 Comparison of emission factors used for Domestic sector in this study and in REAS v2.1**

| (unit: kg/ TJ) | Diesel | Heavy oil | Kerosene | *Oil (REAS)* | LPG | Natural gas | *Gas (REAS)* |
|---|---|---|---|---|---|---|---|
| CO | - | - | 167.57 | *348.00* | 78.65 | - | *77.30* |
| NOx | - | - | 24.94 | *93.20* | 37.21 | - | *61.00* |
| SO2 | - | - | 0.57 | *197.00* | 6.98 | - | *0.24* |
| CH4 | - | - | 2.04 | - | 2.96 | - | - |
| PM | - | - | 43.08 | *4.18* | 5.50 | - | *0.00* |
| NMVOC | - | - | 8.84 | *44.40* | 33.83 | - | *5.00* |
| BC | - | - | 20.41 | *0.55* | 4.23 | - | *0* |



| | | | | | | | |
|---|---|---|---|---|---|---|---|
| OC | - | - | 2.04 | *0.33* | 1.06 | - | *0* |
| NH3 | - | - | - | - | - | - | - |
| N2O | - | - | 1.59 | - | 1.90 | - | - |
| CO2 | - | - | 70975.06 | - | 63002.11 | - | - |