# Peer review of "Technical note: Emission mapping of key sectors in Ho Chi Minh city, Vietnam using satellite derived urban land-use data."

_Atmospheric Chemistry and Physics, 2020_

## Referee Comment (RC1) · Beatriz Aristizabal (Referee) · 26 Oct 2020

Overall quality, general comments

The paper integrates a number of tools to allocate EIs, I consider this is an important part and can be different from the other methodologies reported until now. However, the methodology does not show clearly how the tools are integrated to generate better results. The abstract, introduction, and results analysis focus on the EI. These parts do not show consistently the two principal parts of the paper the EI for the Ho Chi Minh City, but also the novel methodology that the authors implemented. I consider the authors should be careful with some statements and expressions that are not recom-

mended in technical papers. Please see the specific observations below.

Individual scientific questions/issues ("specific comments")

-The title of the paper and the abstract relate a novel method to allocate EI. Since this is an important part of the paper, the abstract should mention how is the novel method developed (give an idea of the main parts). The abstract is focused on the local EI for the HCMC, it would be important to mention the two main points of the paper the local EI but also the novel method. - In the introduction, the authors should explain deeper and in context with other methodologies (that already exist) the novel approach presented in this work, which is the main difference. For example, other methodologies focus only on the disaggregation of transport emissions. It is not enough to mention that they developed a novel approach. The introduction might have more about this aspect. Also, the introduction is very focused on the EI from the city, but don't show the relevance of this study in comparison with other studies that propose allocation or disaggregation methodologies. The authors should focus most on that, the title mention "using satellite..." but there is no information about the importance of these in the context of allocating EI that can help other cities to use this methodology. -The authors mention (around 35) "Many atmospheric chemistry modelling researches in Asia have applied these EIs as input data but they are incoherent and not longer updated". Why are that EI incoherent? the authors should be more specific. Also, I saw in Table 1, there is an updated inventory from 2019. In general, inventories depend on many factors, and determine which one is correct is not easy, how the authors establish which inventory is correct? - Please, include a figure that describes better the methodology for the spatial allocation (disaggregation), because that is the novel part. The authors can improve Figure 1 and 2, or include a new figure focus on the allocation part. Figures 1 and 2, only showed a box mention spatial allocation and the resolution. The methodology for spatial allocation is not available. I also consider Figure 1 and 2 should be improved. -Methodology. I consider some equations are without references, please check these. For example, Ec 1 and 2, they are taking from specific methods

to calculate EIs that are well known. -The authors mention (around 170) "Fuel consumption in 2013, 2014, 2015 were provided by GHG emission inventory compiled by JICA, 2017 (Tab. 4) The fuel consumptions in other years were inferred using population provided by HCMC Statistical Yearbook (Tab.8)". Which correlation/statistic do the authors use for the years 2009, 2010, etc? How the information is "inferred"? The authors again mention "So, electricity consumptions in other years were inferred using the same parameters used in Fuel consumption part". I consider the authors should establish clear the correlation (which parameter? this is not mentioned) because when EIs are calculated this information can affect the results. -Section 2.3.5 Spatial allocation, the authors explained how to allocate EI for each source: transport, point sources (residential, industrial, commercial), however, it is not clear how the total EI is integrated. Also, it is not clear which is the step in the integration of urban morphological maps, DSM, DEM in the case of point sources. Additionally, the authors stated "The composite Landsat (Landsat 7 for 4 years: 2009 to 2012 and Landsat 8 for other four years: 2013 to 2016) was classified in a supervised manner using Mahalanobis distance into 7 classes (including class built-up)", which are the 7 classes used? Since this is the novel methodology, I consider this should be presented in a more organized way. -In the last part of the section Summary and discussions (around line 515), the authors said "We relied on only one building height data (extracted from AW3D30) in 2011 to prepare land use maps for 8 years. The assumption of constant building height neglects vertical growth and land-use transitions, causing inevitable uncertainty in the spatial allocation of emission. Also, this approach assumes that all constituents of the field data of building height and land use could improve the reliability of annual urban morphology maps". That details should be in the methodology about the approximation to use land maps, etc. Additionally, how do the authors "assume" that? - For the sections Results, conclusions, I would recommend an analysis that clearly establishes how the new methodology makes a difference from the previous EIs available for the city. For example, if the transport is the main source as the author mentioned and the methodology implemented in the present work is similar for the spatial disaggregation of transport, how the allocation of point sources using satellite for urban land-use morphological maps improved the resolution and the information (comparing with the previous study of "H.Q.Bang, 2010, the first emission maps were developed for HCMC using road network as allocation factor for Transportation sector, population density as allocation factor for Industrial and Residential sectors").

Technical corrections

-Please review the upper case letters in all the document. I recommend use only when it is strictly necessary. For example, the words Green house gases (in the abstract); transportation, manufacturing, and residential are written in different ways sometimes the author used upper case others no. Please check these typing errors in all the document. Also the space between words, for example, Scope 1 or Scope1 (to standardize). -Check informal English, line 40 "till". - Avoid expressions such as "obviously", I consider this is a qualitative judgment, the authors should show quantitative ideas and support why the approaches are not suitable with precise information (see around line 45) -Some paragraph has unclear phrases or incomplete sentences. For example, around lines 70 and 75. -Please check consistency between plural and singular. For example The daily VKT of each vehicle type in HCMC, 2013 were extracted from study of N.T.K. Oanh, 2015 and was assumed to be constant over years.

---

## Referee Comment (RC2) · Anonymous Referee #2 · 23 Nov 2020

General comments

The paper presents a novel method to generate an updated spatial emission inventory (EI) at city scale using remote sensing data. A high quality EI, of course, is crucial for air quality modeling and designing mitigating strategies. However, the authors do not show clearly how their new method was developed to generate better results. In general, the emphasis of this paper was given mostly to the show the EI results of Ho Chi Minh City, rather than the major differences and advantages of their new methodology compared with other existing methods. As a technical note, the descriptions of the novel method were not complete and precise enough to allow their reproduction by fellow scientist.

[Figure]

The lack of an uncertainty analysis that quantity the errors of the emission inventory is also a major concern. I would expect to see such simulations performed as part of this study, especially for their novel allocation method.

Specific comments

Abstract: The authors should talk more about the novel methodology they use, rather than the EI results of the city. Introduction Line 44: Please remove Taiwan from the list. It is NOT a country. The authors compare the difference among previous EIs, which is quite helpful. But they should also talk more about the major improvements of their novel methodology in comparison with other studies that use different allocation methodologies. Higher detail level of activity data, local emission factors and a novel approach for grid allocation are used in this study. It is obvious the last one is their major originality. They should focus most on this point and provide more information. For example, what are the advantages of using satellite derived urban land-use morphological maps? Table 1: The table should also include a comparison of the time resolutions of these EIs. As mentioned in Introduction, the time resolution of previous EIs can be one year or one month. Why annual emissions (a coarse resolution) are estimated in this study if emissions exhibit strong seasonality? Methodology There is no need to talk about Hanoi. Local emissions are important in all big cities even they are greatly influenced by adjacent sources. Figure 1: The boundary of Ho Chi Minh City is not very clear in the right figure. As mentioned in Introduction, a study by B.Q.Ho et al.., 2019 (Line 69-72) calculate emissions of many pollutants in Ho Chi Minh City in 2017 and also allocate emissions from area sources to grid cells. Why the authors choose 2009 to 2016 as their target year? What is the difference between these two studies? Figure 2 is not clear enough to show how spatial allocation is achieved. A detailed figure that illustrate the complete spatial allocation process is needed in this part. Line 123: Daily VKT can be influenced by many factors. Please justify the assumption that VKT is constant over years. The spatial allocation results should be validated by field measurement data. Results Discussions Line 394: A Study of N.T.K.Oanh et al,

2015 applied the same method with this study. Does it mean the methodology used in this study is not a novel one? Line 440-441: why not use the same year to compare the results between two versions? I think it is more useful to show the improvements of the novel method. As many assumptions and average values are used in this study (for example Line 125-130, Line 157-160 and elsewhere), the authors should try their best to justify these assumptions and discuss the uncertainties associated with these assumptions and average values. I recommend a Monte Carlo simulation or other methods to be used in this study to quantify the uncertainty of each estimation process and the overall uncertainty.

Technical corrections

Please check the use of subscripts throughout the manuscript. For example, CO2 should be $CO_2$. Also use the right significant figures and proper units in all the Tables. The numbers present in the tables contain too many digits and hard to read. Punctuations are missing in some sentences. For example in Line 394.

---

## Author Comment (AC1) · 8 Jan 2021

The authors would like to thank Referee #1 and Anonymous Referee #2 for taking their time to review our manuscript and for giving very constructive and informative comments. These comments helped us improve the quality and clarity of the manuscript. We revised our manuscript based on them. Below are our responses to each comment.

The structure of this document is as follows:

(1) Comments, author's response, and author's changes in manuscript related to Referee #1

(2) The revised main manuscript where changed parts were yellow highlighted

**Author's response, and author's changes in manuscript related to Referee #1:**

*1. The abstract should mention how is the novel method developed (give an idea of the main parts). The abstract is focused on the local EI for the HCMC, it would be important to mention the two main points of the paper the local EI but also the novel method.*

We revised the abstract with more information about the novel method: "Our originality is the use of satellite derived urban land-use morphological maps which allow spatial disaggregation of emissions. We investigated the possibility of using freely available coarse resolution satellite derived digital surface model (DSM) to estimate building height. Building height is combined with urban built-up area classified from Landsat images and nighttime light data to generate annual urban morphological maps. With outstanding advantages of these remote sensing data, our novel method is expected to make a major improvement in comparison with conventional allocation methodologies such as basing on population data."

*2. In the introduction, the authors should explain deeper and in context with other methodologies (that already exist) the novel approach presented in this work, which is the main difference. For example, other methodologies focus only on the disaggregation of transport emissions. It is not enough to mention that they developed a novel approach. The introduction might have more about this aspect.*

We revised the introduction with more information about our novel approach: "With respect to grid allocation, spatial distribution of emissions is a crucial step to fulfil the requirements of gridded EI as input data for air quality modelling. Top-down EIs are often being used as input data for modelling activities at urban scale after downscaling (Susana López-Aparicio et al., 2017). In conventional way, other methodologies focus only on the disaggregation of transport emissions using traffic counts and road network data (C.D. Gómez et al, 2018). The spatial allocation of area source emissions is mainly based on rural, urban and total population data (J. Kurokawa et al, 2013; J. Kurokawa et al, 2019). These approaches are not suitable for community scale EIs what demand higher detail levels of both activity data and spatial disaggregation. Especially, it is not rational to use population data for spatial disaggregation of Industrial sector. Using these methodologies without consideration could lead to underestimation of emissions in urban centre, industrial zones, as well as overestimation in residential zones (P.Saide et al, 2009). It is worth mentioning that these spatial proxies have a strong influence on simulations of air quality modelling, especially when the results are considered for policy making and planning options (M. Trombetti et al, 2018). J Kühlwein et al., 2002 made comparisons among spatial distribution of EIs computed with different levels of information and concluded that a big source of uncertainty is encountered when only considering disaggregation using population. M. Trombetti et al, 2018 also conducted an inter-comparison of the main top-down EIs currently used for air quality modelling studies at the European level regarding downscaling approaches and choice of spatial proxies. Their finding is that the traditional proxies used for gridding residential emissions (e.g. population density) would not be any more relevant. A few studies used land use map as a proxy for deriving spatial patterns of emissions (P.Saide et al, 2009). Nowadays, remote-sensing information is quite important source for land use/land cover modelling. The most prominent advantages of satellite images influencing the spatial allocation of emissions are the ability to collect information over large spatial areas and the ability to collect imagery

of the same area of the earth's surface at different periods in time. By imaging on a continuous basis at different times it is possible to monitor the changes in land use in community scale if the resolution of data is high enough. Moreover, data collected through remote sensing is analyzed at the laboratory which minimizes the work that needs to be done on the field. Accordingly, in the context of spatial allocating emission at a finer scale, remote sensing data is quite promising approach that allows repetitive land use mapping in different study areas. In the case of HCMC, only a few attempts have been made to spatial disaggregate the emissions. Applying similar method with previous works, study of B.Q.Ho, 2010 provides the first emission maps for HCMC using road network as allocation factor for Transportation sector and population density as allocation factor for Industrial and Residential sectors."

*3. The introduction is very focused on the EI from the city, but don't show the relevance of this study in comparison with other studies that propose allocation or disaggregation methodologies.*
In the introduction, we explained that: "In the case of HCMC, only a few attempts have been made to spatial disaggregate the emissions. Applying similar method with previous works, study of B.Q.Ho, 2010 provides the first emission maps for HCMC using road network as allocation factor for Transportation sector and population density as allocation factor for Industrial and Residential sectors."

*4. The title mention "using satellite..." but there is no information about the importance of these in the context of allocating EI that can help other cities to use this methodology.*
We added this part in the introduction: "Nowadays, remote-sensing information is quite important source for land use/land cover modelling. The most prominent advantages of satellite images influencing the spatial allocation of emissions are the ability to collect information over large spatial areas and the ability to collect imagery of the same area of the earth's surface at different periods in time. By imaging on a continuous basis at different times it is possible to monitor the changes in land use in community scale if the resolution of data is high enough. Moreover, data collected through remote sensing is analyzed at the laboratory which minimizes the work that needs to be done on the field. Accordingly, in the context of spatial allocating emission at a finer scale, remote sensing data is quite promising approach that allows repetitive land use mapping in different study areas."

*5. The authors mention (around 35) "Many atmospheric chemistry modelling researches in Asia have applied these EIs as input data but they are incoherent and not longer updated". Why are that EI incoherent? the authors should be more specific.*
We realized that "incoherent" is improper word here, so this part was omitted already.

*6. In Table 1, there is an updated inventory from 2019. In general, inventories depend on many factors, and determine which one is correct is not easy, how the authors establish which inventory is correct?*
To determine which inventory is correct or how good the emission inventory is, the most common approach is using simulation models. For our next step, we plan to apply the EI calculated in this study as input of an air dispersion model to evaluate its reliability. In this study, we chose REASv2.1 as predecessor one. Because while other countries like China and Japan have their own EIs, at that moment, REASv2.1 is the only comprehensive one covering Vietnam. Previous studies had thus applied REASv2.1 in air quality simulations over Vietnam and big cities in Vietnam.

*7. Please, include a figure that describes better the methodology for the spatial allocation (disaggregation), because that is the novel part. The authors can improve Figure 1 and 2, or include a new figure focus on the allocation part. Figures 1 and 2, only showed a box mention spatial allocation and the resolution. The methodology for spatial allocation is not available. I also consider Figure 1 and 2 should be improved.*

We added a Figure about spatial allocation in Figure 2:

[Figure]

*8. Methodology. I consider some equations are without references, please check these. For example, Eq. 1 and 2, they are taking from specific methods to calculate EIs that are well known.*

We added the citations for equations:

$E_{hot} = \sum_i VP_i * DailyVKT_i * 365 * EF_i$  (1) (Creutzig et al., 2011)

$E_{cold} = E_{hot} * \beta_i * F_i$ (2) (Ahlvik P.et al, 1997)

$E_{Fuel} = \sum_i A_i * EF_i * (1 - R_i)$ (3) (IPCC, 1996)

$EF_{SO2} = S_i * (1 - SR_i)$ (4) (IPCC, 1996)

*9. The authors mention (around 170) "Fuel consumption in 2013, 2014, 2015 were provided by GHG emission inventory compiled by JICA, 2017 (Tab. 4) The fuel consumptions in other years were inferred using population provided by HCMC Statistical Yearbook (Tab.8)". Which correlation/statistic do the authors use for the years 2009, 2010, etc? How the information is "inferred"? The authors again mention "So, electricity consumptions in other years were inferred using the same parameters used in Fuel consumption part". I consider the authors should establish clear the correlation (which parameter? this is not mentioned) because when EIs are calculated this information can affect the results.*

We wrote more for the explanation of this part in Methodology: "The fuel consumptions of Manufacturing industrial sector in five other years (2009 to 2012 and 2016) were proportional calculated using annual Gross output of industry at current prices by industry activity in HCMC, provided by HCMC Statistical Yearbook (Tab.8) with the assumption that there is linear relationship between Fuel consumption of Manufacturing industrial sector and Gross output of industry"

And

"Fuel consumption in 2013, 2014, 2015 were provided by GHG EI compiled by JICA, 2017 (Tab. 6) The fuel consumptions in other years were proportional calculated using population provided by HCMC Statistical Yearbook (Tab.8) with the assumption that there is linear relationship between Fuel consumption of Residential sector and population of city."

And

"electricity consumptions in other years were proportional calculated using the same proxies applied in Fuel consumption part:
-        Manufacturing Industries and Construction sector:   used Annual Gross output of industry at current prices by industry activity with the assumption that there is linear relationship between electricity consumption of Manufacturing industrial sector and Gross output of industry.
-        Residential: used Annual population of HCMC with the assumption that there is linear relationship between electricity consumption of Residential sector and population of city."

*10. Section 2.3.5 Spatial allocation, the authors explained how to allocate EI for each source: transport, point sources (residential, industrial, commercial), however, it is not clear how the total EI is integrated.*
We wrote more for the explanation of this part in Methodology: "These land use maps were used for spatial distribution of Manufacturing industrial and Residential emissions into the same grid net – 1 km resolution with Transportation sector. Basing on the spatial matching, total emission of three key sectors is simply sum of grid nets of Manufacturing industrial, Residential and Transportation emissions."

*11. It is not clear which is the step in the integration of urban morphological maps, DSM, DEM in the case of point sources.*
We explained in the Spatial allocation part: "Noteworthy, in this study industrial emission sector is considered as area source instead of point source like previous studies."

*12. Additionally, the authors stated "The composite Landsat (Landsat 7 for 4 years: 2009 to 2012 and Landsat 8 for other four years: 2013 to 2016) was classified in a supervised manner using Mahalanobis distance into 7 classes (including class built-up)", which are the 7 classes used? Since this is the novel methodology, I consider this should be presented in a more organized way.*
We revised this part with more detailed explanations: "A time-series of Landsat imagery (Landsat 7 for 4 years: 2009 to 2012 and Landsat 8 for other four years: 2013 to 2016) was classified to generate the urban built-up extent for 2009 to 2016. A Mahalanobis distance based supervised classification was performed to identify 5 classes (including built up, vegetation, fallow land, lake and river)."

*13. In the last part of the section Summary and discussions (around line 515), the authors said "We relied on only one building height data (extracted from AW3D30) in 2011 to prepare land use maps for 8 years. The assumption of constant building height neglects vertical growth and land-use transitions, causing inevitable uncertainty in the spatial allocation of emission. Also, this approach assumes that all constituents of the field data of building height and land use could improve the reliability of annual urban morphology maps". That details should be in the methodology about the approximation to use land maps, etc. Additionally, how do the authors "assume" that?*
The objective of urban morphology mapping is to classify annual land use into three classes: residential, commercial and industrial. The assumption of constant building height is valid as long as the change in building heights does not impact the transitions among those three land use types. Our simple random resampling in Google earth proved that the land use change occurring in HCMC mainly is urban expansion (95%) The transitions among residential- commercial – industrial barely happen. So neglecting vertical growth of city will cause inevitable uncertainty in the spatial allocation of emission.

*14. For the sections Results, conclusions, I would recommend an analysis that clearly establishes how the new methodology makes a difference from the previous EIs available for the city. For example, if the transport is the main source as the author mentioned and the methodology implemented in the present work is similar for the spatial disaggregation of transport, how the allocation of point sources using satellite for urban land-use morphological maps improved the resolution and the information (comparing*

*with the previous study of "H.Q.Bang, 2010, the first emission maps were developed for HCMC using road network as allocation factor for Transportation sector, population density as allocation factor for Industrial and Residential sectors")*

We wrote more explanations for this in Conclusion part: "The previous emission maps available for HCMC include works of B.Q.Ho, 2010, B.Q.Ho et al, 2019 and REAS inventories. Those studies used road network and population density data for spatial allocation. The novelty of this study is that the disaggregation of transportation emission based on road density combining with different weights for three types of roads that are their traffic volumes, while residential and commercial sectors were allocated by urban morphology maps. In conventional way, field-work based data are labor consuming and cannot be performed frequently. Also, the use of those existing spatial distribution surrogates neglects the effects of urban sprawl that is evident in big cities. It is desirable to have access to revised spatial allocation factors that may be more representative of spatial distributions in community scale and more available. And even if statistical data is inaccessible in other cities remote sensing data can be used. Remote sensing data can be updated frequently, too. Thus, the use of satellite images makes spatial disaggregation updating quite simple and efficient. Besides, it is the best tool to represent urban expansion and land use change, so it ensures the accuracy of grid allocation when closely related spatial activity surrogate is needed to compile EI in local scale"

A comprehensive analysis of the improvement of our EIs will be conducted in our future study that uses these EIs as input data of atmospheric chemistry models and conduct the comparison to independently derived data.

*15. Technical corrections:*
We revised the manuscript according to these corrections as shown in *The revised main manuscript.*

**The revised main manuscript where changed parts were yellow highlighted:**

[revised manuscript text omitted]
 **±19** | **±24** | **±22** | **±23** | **±31** | | **±25** | **±31** | **±34** | **±27** | **±31** | **±20** |

---

## Author Comment (AC2) · 8 Jan 2021

The authors would like to thank Referee #1 and Anonymous Referee #2 for taking their time to review our manuscript and for giving very constructive and informative comments. These comments helped us improve the quality and clarity of the manuscript. We revised our manuscript based on them. Below are our responses to each comment.

The structure of this document is as follows:

- (1) Comments, author's response, and author's changes in manuscript related to Referee #2
- (2) The revised main manuscript where changed parts were yellow highlighted

**Author's response, and author's changes in manuscript related to Referee 2:**

1. Abstract: The authors should talk more about the novel methodology they use, rather than the EI results of the city.

We revised the abstract with more information about the novel method: "Our originality is the use of satellite derived urban land-use morphological maps which allow spatial disaggregation of emissions. We investigated the possibility of using freely available coarse resolution satellite derived digital surface model (DSM) to estimate building height. Building height is combined with urban built-up area classified from Landsat images and nighttime light data to generate annual urban morphological maps. With outstanding advantages of these remote sensing data, our novel method is expected to make a major improvement in comparison with conventional allocation methodologies such as basing on population data."

2. Introduction Line 44: Please remove Taiwan from the list. It is NOT a country. We omitted Taiwan in this line.

3. The authors compare the difference among previous EIs, which is quite helpful. But they should also talk more about the major improvements of their novel methodology in comparison with other studies that use different allocation methodologies. Higher detail level of activity data, local emission factors and a novel approach for grid allocation are used in this study. It is obvious the last one is their major originality. They should focus most on this point and provide more information. For example, what are the advantages of using satellite derived urban land-use morphological maps?

We added more information about the novel methodology and the advantages of using satellite derived urban land-use morphological maps in the Introduction: "With respect to grid allocation, spatial distribution of emissions is a crucial step to fulfil the requirements of gridded EI as input data for air quality modelling. Top-down EIs are often being used as input data for modelling activities at urban scale after downscaling (Susana López-Aparicio et al., 2017) In conventional way, other methodologies focus only on the disaggregation of transport emissions using traffic counts and road network data (C.D. Gómez et al, 2018). The spatial allocation of area source emissions is mainly based on rural, urban and total population data (J. Kurokawa et al, 2013; J. Kurokawa et al, 2019) These approaches are not suitable for community scale EIs what demand higher detail levels of both activity data and spatial disaggregation. Especially, it is not rational to use population data for spatial disaggregation of Industrial sector. Using these methodologies without consideration could lead to underestimation of emissions in urban centre, industrial zones, as well as overestimation in residential zones (P.Saide et al, 2009). It is worth mentioning that these spatial proxies have a strong influence on simulations of air quality modelling, especially when the results are considered for policy making and planning options (M. Trombetti et al, 2018). J Kühlwein et al., 2002 made comparisons among spatial distribution of Els computed with different levels of information and concluded that a big source of uncertainty is encountered when only considering disaggregation using population. M. Trombetti et al, 2018 also conducted an inter-comparison of the main top-down EIs currently used for air quality modelling studies at the European level regarding downscaling approaches and choice of spatial proxies. Their finding is that the traditional proxies used for gridding residential emissions (e.g. population density) would not be any more relevant. A few studies used land use map as a proxy for deriving spatial patterns of emissions (P.Saide et al, 2009) Nowadays, remotesensing information is quite important source for land use/land cover modelling. The most prominent advantages of satellite images influencing the spatial allocation of emissions are the ability to collect information over large spatial areas and the ability to collect imagery of the same area of the earth's surface at different periods in time. By imaging on a continuous basis at different times it is possible to monitor the changes in land use in community scale if the resolution of data is high enough. Moreover, data collected through remote sensing is analyzed at the laboratory which minimizes the work that needs to be done on the field. Accordingly, in the context of spatial allocating emission at a finer scale, remote sensing data is quite promising approach that allows repetitive land use mapping in different study areas. In the case of HCMC, only a few attempts have been made to spatial disaggregate the emissions. Applying similar method with previous works, study of B.Q.Ho, 2010 provides the first emission maps for HCMC using road network as allocation factor for Transportation sector and population density as allocation factor for Industrial and Residential sectors."

| Emission
inventories | References        | Species          | Years       | Area covered  | Spatial resolution | Time resolution |
|-------------------------|-------------------|------------------|-------------|---------------|--------------------|-----------------|
|                         | Kato and          | SO2 and NOx      | 1975, 1980, | East Asian,   | 1°×1°              | Annual          |
|                         | Akimoto           |                  | 1985, 1986  | Southeast     |                    |                 |
|                         | (1992)            |                  | and 1987    | Asian and     |                    |                 |
|                         |                   |                  |             | South Asian   |                    |                 |
|                         |                   |                  |             | countries     |                    |                 |
| TRACE-P                 | Jacob et          | CO2, CH4, N2O,   | 2000        | Over western  |                    |                 |
|                         | al. , 2003 | 03, CFC, CO, SO2 |             | Pacific       |                    |                 |
| INTEX-B                 | Zhang et          | CO2, CH4, N2O,   | 2006        | Over western  |                    |                 |
|                         | al., 2009         | O3, CFC, CO, SO2 |             | Pacific       |                    |                 |
| REASv1.1                | T. Ohara et       | SO2, NOx, CO,    | From 1980   | East,         | 0.5°×0.5°          | Monthly         |
|                         | al, 2007          | NMVOC, BC, OC,   | to 2020     | Southeast and |                    |                 |
|                         |                   | CO2, NH3, CH4    |             | South Asia    |                    |                 |
|                         |                   | and N2O          |             |               |                    |                 |
| REASv2.1                | J.                | SO2, NOx, CO,    | From 2000   | East,         | 0.25∘×0.25∘        | Monthly         |
|                         | Kurokawa          | NMVOC, PM2.5,    | to 2008     | Southeast,    |                    |                 |
|                         | et al, 2013       | PM10, BC, OC,    |             | South Asia,   |                    |                 |
|                         |                   | CO2, NH3, CH4    |             | Central Asia  |                    |                 |
|                         |                   | and N2O          |             | and Russia    |                    |                 |
|                         |                   |                  |             | Asia          |                    |                 |
| REASv3.1                | J.                | SO2, NOx, CO,    | During      | East,         | 0.25∘×0.25∘        | Monthly         |
|                         | Kurokawa          | NMVOC, PM2.5,    | 1950-1955   | Southeast,    |                    |                 |
|                         | et al, 2019       | PM10, BC, OC,    | and from    | South Asia,   |                    |                 |
|                         |                   | CO2, NH3, CH4    | 2010-2015   | Central Asia  |                    |                 |
|                         |                   | and N2O          |             | and Russia    |                    |                 |
|                         |                   |                  |             | Asia          |                    |                 |
| MIX                     | Tsinghua          | SO2, NOx, CO,    | 2008 and    | East,         | 0.25°×0.25°        | Monthly         |
|                         | Universitv        | NMVOC, NH3,      | 2010        | Southeast.    |                    |                 |

4. Table 1: The table should also include a comparison of the time resolutions of these Els. We revised Table 1 according to this comment:

| (Zhang et          | PM10, PM2.5,   | South Asia,  |
|--------------------|----------------|--------------|
| al., 2009;         | BC, OC and CO2 | Central Asia |
| Li et al.,         |                | and Russia   |
| 2014;              |                | Asia         |
| Zheng et           |                |              |
| al. , 2014) |                |              |

5. As mentioned in Introduction, the time resolution of previous Els can be one year or one month. Why annual emissions (a coarse resolution) are estimated in this study if emissions exhibit strong seasonality? In this study, we estimated annual emissions in Ho Chi Minh city because the emissions here do not exhibit strong seasonality as ones in Hanoi. Besides, HCMC is located in tropical region, so the significant monthly variations in fuel consumption and electricity consumption of stationary energy sectors is not expected.

6. Methodology: There is no need to talk about Hanoi. Local emissions are important in all big cities even they are greatly influenced by adjacent sources.

We omitted Hanoi part and revised this part: "HCMC is the most populous city in Vietnam with a population of 9 million as of 2019. Air quality in this city is mainly influenced by anthropogenic emission occurring inside the city. In other words, the relative independence of situation in this city on other adjacent sources facilitates the compiling local EI. Also, the local emissions are dominant sources of pollution (GreenID, 2018)"

7. Figure 1: The boundary of Ho Chi Minh City is not very clear in the right figure. We revised Figure 1 according to this comment:

Figure 1. Ho Chi Minh city – inventory domain of our El (© OpenStreetMap contributors 2019. Distributed under a Creative Commons BY-SA License.)

8. As mentioned in Introduction, a study by B.Q.Ho et al., 2019 (Line 69-72) calculate emissions of many pollutants in Ho Chi Minh City in 2017 and also allocate emissions from area sources to grid cells. Why the authors choose 2009 to 2016 as their target year? What is the difference between these two studies?

We chose 2009 to 2016 as target year to continue the REASv2.1, which is the first inventory to integrate time series of emission data for Asia on the basis of a consistent methodology. A study by B.Q.Ho et al., 2019 estimated emissions in Ho Chi Minh city in a single year – 2017 and they used road network and population density data as proxies for spatial allocation.

9. Figure 2 is not clear enough to show how spatial allocation is achieved. A detailed figure that illustrate the complete spatial allocation process is needed in this part. We added a Figure about spatial allocation in Figure 2:

10. Line 123: Daily VKT can be influenced by many factors. Please justify the assumption that VKT is constant over years.

First of all, although VKT in this study is assumed to be constant, its uncertainty is discussed in Sect. 3.5 with Monte Carlo simulation. Secondly, because the traffic situation in Ho Chi Minh city is specific with the domination of motorcycles, it poses a great challenge to adopt VKTs from previous research conducted in other cities. More deeper studies are needed to consider the impact of urban expansion on the annual changes in VKTs of all vehicle types in Ho Chi Minh city.

**11. The spatial allocation results should be validated by field measurement data.**

For accuracy assessment, we chose 100 random samples from Google earth image in 2016, including 60 samples for Residential area, 20 samples for Industry and 20 samples for Commercial area. The total accuracy is 77%. The user's accuracy and producer's accuracy of Residential class are 88% and 75% respectively. The user's accuracy and producer's accuracy of Industrial class are 58% and 67% respectively. Meanwhile, Commercial area shown 57% and 80% for user's accuracy and producer's accuracy respectively.

**12. Results Discussions Line 394: A Study of N.T.K.Oanh et al, 2015 applied the same method with this study. Does it mean the methodology used in this study is not a novel one?**

A Study of N.T.K.Oanh et al, 2015 estimated Transportation emission in Ho Chi Minh city in a single year – 2013 and they did not conduct the spatial allocation of emission.

13. Line 440-441: why not use the same year to compare the results between two versions? I think it is more useful to show the improvements of the novel method. Because our target years (2009-2016) is different from the target year of REASv2.1 (2000-2008)

14. As many assumptions and average values are used in this study (for example Line 125-130, Line 157-160 and elsewhere), the authors should try their best to justify these assumptions and discuss the uncertainties associated with these assumptions and average values. I recommend a Monte Carlo simulation or other methods to be used in this study to quantify the uncertainty of each estimation process and the overall uncertainty.

We added the Uncertainty part using Monte Carlo simulation in Sect. 3.5

15. Technical corrections:

We revised the manuscript according to these corrections as shown in The revised main manuscript.

The revised main manuscript where changed parts were yellow highlighted:

[revised manuscript text omitted]

---

## Referee Report (RR1)

All the criticisms have been adequately addressed in the revision.